# Single-cell analysis identifies dynamic gene expression networks that govern B cell development and transformation

Robin D. Lee [1], Sarah A. Munro[2], Todd P. Knutson [2], Rebecca S. LaRue[2], Lynn M. Heltemes-Harris[1] & Michael A. Farrar [1]✉

Integration of external signals and B-lymphoid transcription factor activities organise B cell lineage commitment through alternating cycles of proliferation and differentiation, producing a diverse repertoire of mature B cells. We use single-cell transcriptomics/proteomics to identify differentially expressed gene networks across B cell development and correlate these networks with subtypes of B cell leukemia. Here we show unique transcriptional signatures that refine the pre-B cell expansion stages into pre-BCR-dependent and pre-BCR-independent proliferative phases. These changes correlate with reciprocal changes in expression of the transcription factor EBF1 and the RNA binding protein YBX3, that are defining features of the pre-BCR-dependent stage. Using pseudotime analysis, we further characterize the expression kinetics of different biological modalities across B cell development, including transcription factors, cytokines, chemokines, and their associated receptors. Our findings demonstrate the underlying heterogeneity of developing B cells and characterise developmental nodes linked to B cell transformation.

[1] Department of Laboratory Medicine and Pathology, Center for Immunology, Masonic Cancer Center, University of Minnesota, Minneapolis, United States. [2] Minnesota Supercomputing Institute, University of Minnesota, Minneapolis, United States. ✉email: farra005@umn.edu

Distinct stages of B-cell development have been delineated using flow cytometry and a variety of surface[1,2] and intracellular markers[3,4]. The use of such markers in combination with distinct gene knockout mice has greatly expanded our understanding of specific B-lymphoid transcription factors[5–7], cytokines[8–10], and signaling pathways that entrain B cell development. However, these markers are insufficient to fully demarcate distinct subsets[1], resulting in the analysis of mixed populations. These limitations have led to an incomplete understanding of B-lymphoid transcription-factor expression kinetics across the B-cell developmental trajectory, and the orchestration of transcriptional programs underlying the alternating cycles of proliferation and differentiation. Capturing transition states as B cells differentiate from one stage to the next is particularly difficult. Furthermore, perturbations during normal B-cell differentiation can lead to development of B-cell acute lymphoblastic leukemia (B-ALL)[11–13]. However, exactly what stages are most permissive for transformation remains imprecisely defined. Recent characterization of B-ALL subtypes showed diverse transcriptional signatures, suggesting multiple points of origin, or use of different signaling pathways to drive transformation[12,14]. Therefore, understanding normal B-cell transcriptional programs can determine where transformation occurs and how B-ALLs exploit B-cell developmental pathways.

To address the above questions, we used single-cell transcriptomics (scRNA-Seq) and proteomics (CITE-Seq, Cellular Indexing of Transcriptomes and Epitopes by Sequencing)[15] to precisely characterize different subsets of B-cell development. Our analysis discovered several stages of pre-B-cell differentiation—including a pre-BCR-dependent and two pre-BCR-independent stages that exhibited distinct modalities of proliferation. This process of pre-B-cell differentiation was characterized by oscillatory regulation of the transcription factor EBF1, with reciprocal changes in the RNA-binding protein YBX3. In contrast, the pre-BCR-independent stages correlated with changes in chemokine and cytokine receptors and suggest that these stages may involve differential localization of pre-BCR-independent stage subsets within the bone marrow. Finally, comparisons of various human B-ALL transcriptomes to those of different stages of B-cell development highlight the developmental nodes that associate with varying subtypes of B-ALL and how they correlate with prognosis.

## Results

### Identifying B-cell development stages using scRNAseq and CITE-Seq.
To couple transcriptional information with B-cell-stage-defining surface marker expressions, we used combined single-cell RNA sequencing and CITE-Seq (also referred as antibody-derived-tags [ADT] hereafter) proteomics. Bone marrow from two wildtype C57BL/6 mice was harvested and stained with two distinct oligo-labeled antibodies that recognize CD45 and MHC class I, which allow identification of cells derived from each individual mouse (referred to as hashtag antibodies). We further stained cells with a panel of CITE-Seq antibodies (B220, CD19, CD93, CD25, IgM, and CD43) as well as fluorescently labeled B220 and CD43. Cells were sorted (representative gating; Supplementary Fig. 1a) at a 1:1 ratio of B220$^+$CD43$^+$ and B220$^+$CD43$^-$ cells to enrich for early progenitor B-cell subsets (Fig. 1a). This sorting scheme captures the vast majority of developing B cells in the bone marrow but does exclude a small fraction of developing CD19$^+$ B cells that express CD11c, Ly6G, or NK1.1 (~1–3%) (Supplementary Fig. 1b). After processing samples, the data set contained 7454 single cells contributing to 14 transcriptionally unique clusters (Fig. 1b). Cell cycle status was determined by measuring the average expression of gene sets

representing canonical S and G2M phases and a recently described postmitotic G1 phase (G1PM) that is associated with rapidly cycling cells that contain carryover G2M transcripts[16,17]. Cells were classified by which gene set was most enriched, while cells lacking any of these three signatures were labeled G0/G1 by default (Fig. 1c, Supplementary Fig. 2a). We found that the proliferating cells (G1PM, S, and G2/M phase) were clustered together away from quiescent cells (G0/G1), suggesting that cycling status was a major source of variance (Fig. 1c). Similarly, Mki67 gene expression levels were high in cells classified as G1PM, S, or G2/M phase (Fig. 1c). Further, we found that CITE-seq provided superior sensitivity compared with the corresponding transcript expression (Fig. 1d). Cells marked CD43$^+$ by CITE-Seq comprised the majority of cycling cells, supporting previous characterizations of cycling progenitor B cells[1]. Both wild-type mice were equally represented in all cell clusters (Supplementary Fig. 2b) and had equal detection of all CITE-seq antibodies (Supplementary Fig. 2c, d). Overall, CITE-seq antibody expression recapitulated flow cytometry-based staging of B-cell development and when coupled with transcriptomic signatures provided the basis for demarcating different stages and transitions during B-cell development.

### Transcriptional signatures of pre–pro B cells and committed pro B cells.
We first used antibody-derived CITE-Seq tags to define B220$^+$CD43$^+$CD19$^-$ pre–pro B cells. These cells expressed early B-lineage-associated genes such as Flt3, Il7r, and Cd79a (Fig. 2a). Pre–pro B cells have also been shown to express genes associated with myeloid lineages, consistent with the observation that they can give rise to myeloid cells as well[18,19]. Indeed, we found that the pre–pro B cells have high expression of myeloid lineage-associated transcription factors such as Runx2, Irf8, and Tcf4, and plasmacytoid dendritic cell markers, such as Bst2; these genes were silenced upon commitment to the B-cell lineage at the pro-B-cell stage (Fig. 2b). We also assessed the expression of previously described EBF1-repressed target genes, including Tyrobp, Clec12a, Cd300a, Cd7, Chdh, and Mycl[20] (Fig. 2c). These target genes were highly expressed in the pre–pro B-cell cluster, while Ebf1-expressing pro-B cells had low or undetectable expression of these genes, further distinguishing the pre–pro B cells from pro-B cells (Fig. 2b). To identify the presence of B cell-biased uncommitted progenitors in this cluster, we used CD93 ADT expression, previously shown to enrich for B cell progenitors[21]. A median split of CD93 expression was performed in this cluster, and the CD93 above-median expressing cells had enrichment of B-cell commitment associated genes, such as Ebf1, Cd24a, and Vpreb1 (Fig. 2d). Pro-B cells, traditionally delineated as c-KIT$^+$ cells[22] had Kit gene expression (Fig. 2e). In addition, pro-B cells expressed the Bcl-2 family gene, Bok, and interferon-stimulated genes, such as Ifitm2 and Ifitm3 (Fig. 2e, Supplementary Data 1) Finally, EBF1 positively regulates the expression of pre-BCR-surrogate light-chain genes, Vpreb1 and Igll1[23]. We found that cycling pro-B cells had high expression of both Vpreb1 and Igll1, compared with the pre–pro B cells that do not express Ebf1 (Fig. 2f). Thus, our analysis confirmed several genes expression patterns associated with the transition from pre–pro-B to pro-B cells, as well as identified highly specific markers of pro-B cells such as Bok.

### Pre-B-cell expansion comprises distinct pre-BCR-dependent and pre-BCR-independent proliferation stages.
Next, we assessed the transcriptional signature of the ADT-B220$^+$CD19$^+$ CD43$^+$CD25$^-$ cells. We refer to these cells as the pre-BCR-dependent proliferation cluster (also referred to pre-BCRd hereafter) (Fig. 3a). Pre-BCR signaling initiates silencing of the surrogate

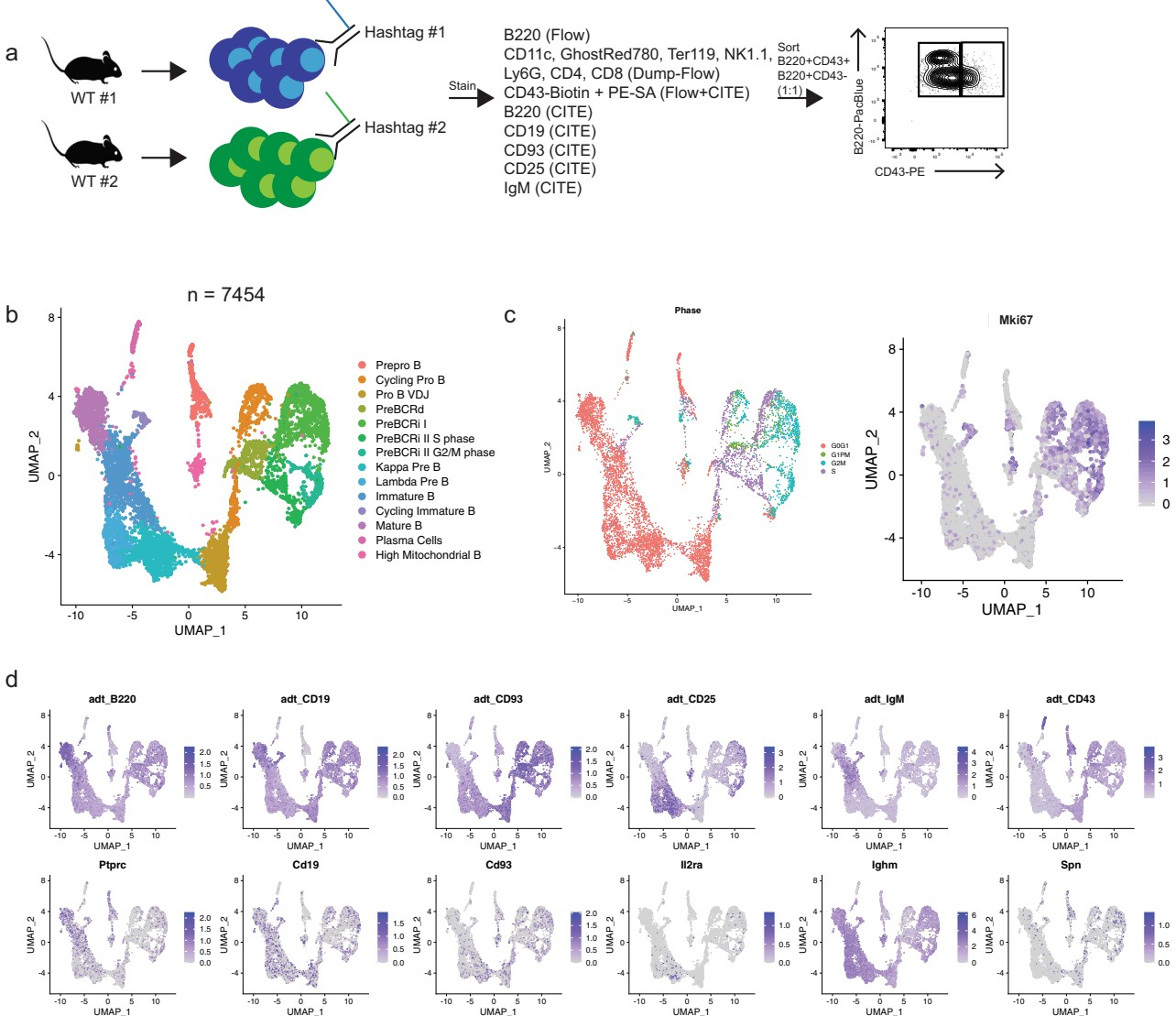

**Fig. 1 Identifying B-cell development stages using scRNAseq and CITE-Seq. a** Schematic of the experimental setup. WT wild type. **b** UMAP dimension-reduction projection of all cells (n = 7454) from two wild-type C57BL6 mice. Fourteen clusters were identified and the corresponding population names for each cluster are listed (Pre-BCRd pre-BCR-dependent, Pre-BCRi pre-BCR-independent) **c** Feature plot of cells that are labeled according to their cell cycle status based on gene expression (left) and *Mki67* transcription expression (right). Cell cycle status was determined by the average expression of gene sets representing each cell cycle, including postmitotic G1 phase (G1PM), S phase, or G2/M phase, and cells that did not harbor any of these signatures were labeled as G0/G1. Color scale represents natural log-transformed SCTransform corrected counts. **d** Feature plot of cells for their CITE-Seq/ADT antibody expression (top row) and the corresponding gene transcript expression (bottom row). Color scale represents centered natural log transformation across cells. adt antibody-derived tags.

light-chain locus[24]. In accordance with this, the pre-BCR-dependent cells have intermediate expression of both *Vpreb1* and *Igll1* (Fig. 2f). Genes that were uniquely upregulated in this cluster included *Nrgn* and *Ybx3* (Fig. 3a, Supplementary Figure 3a and Supplementary Data 2). NRGN (Neurogranin) is a calmodulin-binding protein that regulates the dynamics of calcium binding to calmodulin[25]. Neurogranin is upregulated in activated[26] and anergic B cells[27], which suggests that neurogranin expression is controlled by B-cell receptor signaling. YBX3 is a DNA/RNA-binding protein that was recently shown to stabilize the amino acid transporter transcripts *Slc7a5* and *Slc3a2* in HELA cells, allowing for their robust translation[28]. SLC7A5 and SLC3A2 heterodimerize to form CD98, a large neutral amino acid transporter. Expression of CD98 in CD8+ T cells has been shown to be tightly controlled by antigen-receptor signaling and is critical for MYC expression[29]. Our findings suggest that pre-BCR signaling may serve a similar

function. Consistent with this idea, *Myc* was most highly expressed in the *Ybx3*, *Slc7a5*, and *Slc3a2* expressing pre-BCRd cluster (Fig. 3a). Surprisingly *Ebf1* expression is significantly reduced, while expression of the transcription factor *Pax5* was largely unchanged and *Ikzf1* was modestly induced (Fig. 3b). Concordant with low *Ebf1* expression, EBF1-target genes, such as *Cd79a*[30,31] and *Cd79b*[32], are also significantly reduced in the pre-BCRd stage (Fig. 3c). In addition, *Il7r* expression, a negative regulator of pre-BCR-signaling components[33], is also reduced (Fig. 3c). To identify potential transcriptional regulators that govern this pre-BCR-dependent expansion stage, we performed a Landscape In Silico deletion Analysis (LISA)[34] using the top 100 differentially upregulated genes in the pre-BCRd cluster. After excluding factors not expressed in B cells (i.e., MYCN), we found MYC to be the top predicted regulator of this gene set in pre-BCRd cells. Consistent with its decreased expression, EBF1 was predicted to have minimal

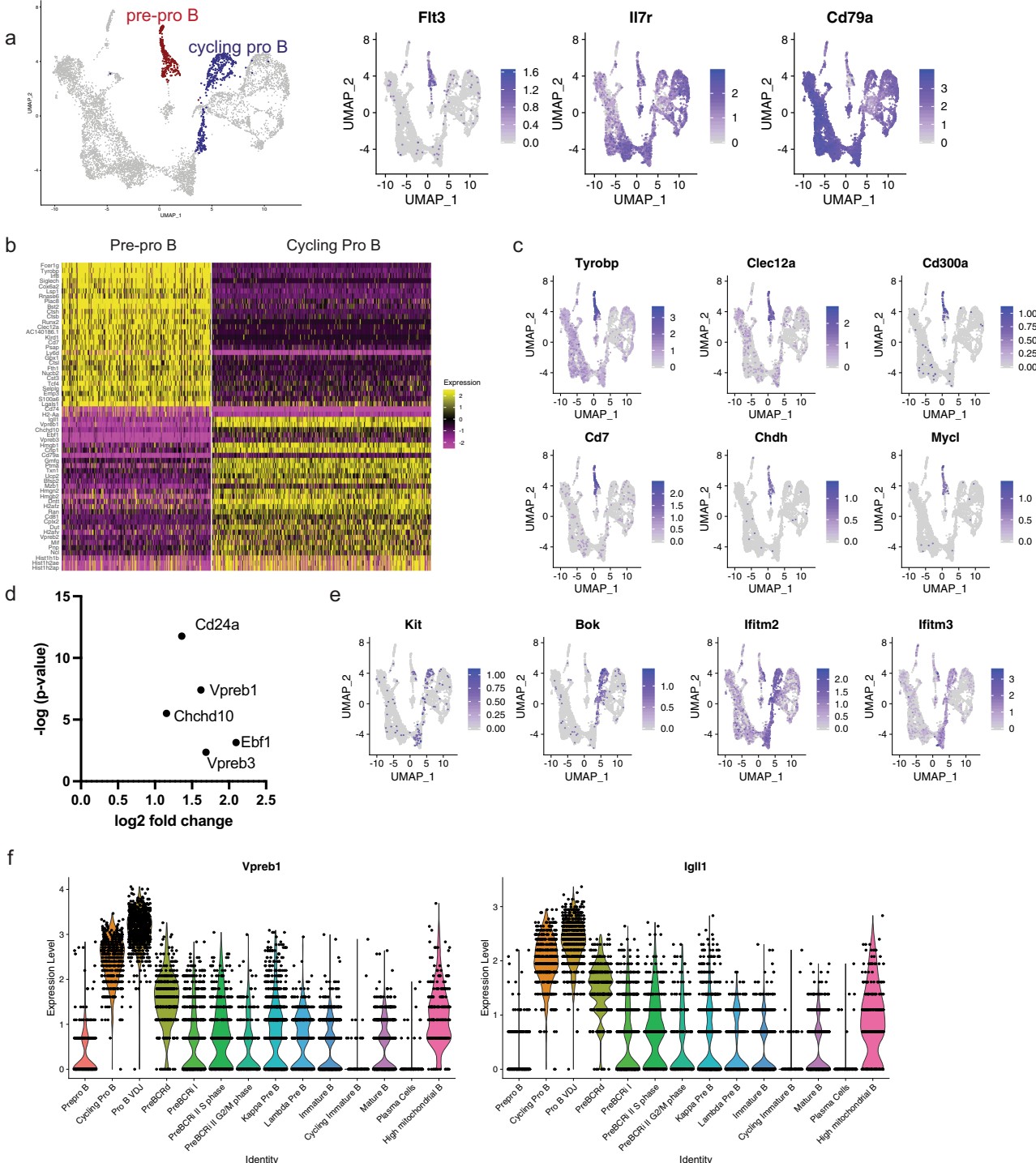

**Fig. 2 Transcriptional signatures of pre–pro B cells and committed pro-B cells. a** Highlighted populations for pre–pro B and cycling pro-B cells (left). Feature plots for early B-cell markers (right). **b** Heatmap of genes that are differentially expressed between the pre–pro B cells and the cycling pro-B cells. Scale represents normalized counts centered and scaled across cells. **c** Feature plot of EBF1-target genes. **d** Differentially upregulated genes in the CD93 ADT high versus CD93 ADT low pre–pro B-cell cluster. CD93 high versus low was based on a median split. No genes were significantly downregulated in the CD93 high population. **e** Feature plot of genes denoting the pro-B-cell stage (*Kit*), or genes uniquely expressed during pro-B cells. **f** Violin plot of the expression of surrogate light chains, *Vpreb1* and *Igll1* across the B-cell development stages. Color scale in a, c, e, and f represents natural log-transformed SCTransform-corrected counts.

contributions in pre-BCRd cells (Fig. 3d). In addition, we performed a gene-set enrichment analysis (GSEA), which further supported a MYC expression signature and enrichment of metabolic reprogramming signatures (Fig. 3d). Together, this suggests that MYC is the critical transcription factor that governs the

transcriptional landscape during pre-BCR-dependent expansion. To further understand the importance of *Ebf1* downregulation during pre-BCR signaling, we examined differentially regulated genes that contained EBF1 binding sites. EBF1 has clear binding sites at the promoters for *Nrgn* and *Myc* (Fig. 3e). Likewise, EBF1 binds to a

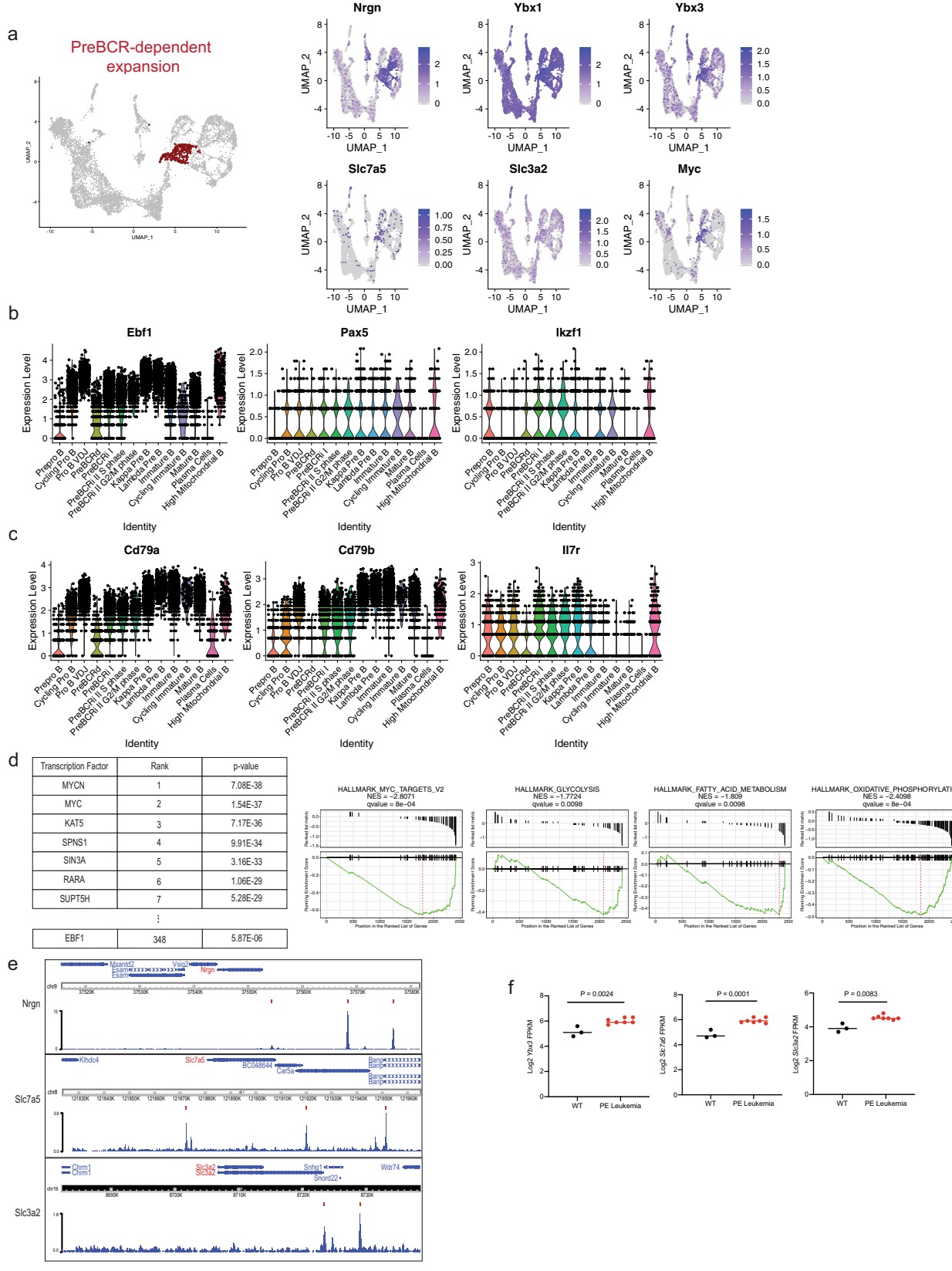

known superenhancer linked to the *Myc* locus[33]. In addition, potential binding sites were observed within the promoters of the *Slc7a5* and *Slc3a2* genes (Fig. 3e). We performed RNA-seq in wildtype and *Pax5$^{+/-}$* × *Ebf1$^{+/-}$* leukemic progenitor B cells and evaluated expression changes of *Ybx3*, *Slc7a5*, and *Slc3a2*. We found that *Slc7a5* and *Slc3a2*, along with other pre-BCR-dependent stage

module genes (*Ybx3*), were upregulated with decreased *Ebf1* gene dosage (Fig. 3f). This is consistent with our recent observation that *Myc* expression is increased in *Pax5$^{+/-}$* × *Ebf1$^{+/-}$* preleukemic and *Pax5$^{+/-}$* × Ebf1$^{+/-}$* leukemic progenitor B cells[35]. These findings suggest that EBF1 mediates repression of the pre-BCR gene expression module.

**Fig. 3 The transcriptome of the pre-BCR-dependent expansion stage is governed by MYC-associated gene expression networks and requires repression of *Ebf1* expression. a** Highlighted population of the pre-BCR-dependent cluster (left) and feature plots for the pre-BCR-dependent markers highly expressed during this stage (right). **b** Violin plot of B-lymphoid transcription-factor expression, including *Ebf1*, *Pax5*, and *Ikzf1* across B-cell development. **c** Violin plot of additional genes that are downregulated during pre-BCR-dependent proliferation, which includes *Il7r*, and EBF1-target genes such as *Cd79a* and *Cd79b*. **d** Landscape In silico Deletion Analysis (LISA) to predict the transcriptional regulator of the top 100 differentially upregulated genes during pre-BCR-dependent proliferation (left table). Gene-set enrichment analysis was performed to identify gene sets from the Molecular Signature Database that were enriched in the pre-BCR-dependent cells. Comparison between the pre-BCR-independent cells (left side) and the pre-BCR-dependent cells was performed (right side). Statistical significance was calculated using one-sided Wilcoxon rank-sum test. **e** EBF1-binding sites at the promoters of *Nrgn*, *Slc7a5* and *Slc3a2* with indicated MACS peak calls[20]. Data were obtained from GSM2863146. **f** Log2-transformed FPKM expression values obtained from RNA-seq of wild-type and Pax5$^{+/-}$ × Ebf1$^{+/-}$ (PE) leukemic progenitor B cells. Progenitor B cells were obtained via negative selection of CD11c, TER119, Ly6G, Ig Kappa, and Ig Lambda and positive selection of CD19. Statistical significance was determined using a two-tailed unpaired student *t*-test for *Ybx3* (*P* = 0.0024) and *Slc7a5* (*P* = 0.0001). RNA-seq data were obtained from GSE148680. A two-tailed Mann–Whitney test was performed for *Slc3a2* (*P* = 0.0083) due to non-normal distribution. *n* = 7 biologically independent samples over one independent experiment. Color scales in **a**, **b**, and **c** represent natural log-transformed SCTransform-corrected counts.

YBX3 binds to *Jak1* transcripts and inhibits translation of *Jak1* in HELA cells[28]. To evaluate whether YBX3 levels correlate with reduced JAK1 in pre-B cells, and to validate EBF1 downregulation during pre-BCR signaling, we used flow cytometry to characterize B220$^+$CD19$^+$CD43$^+$CD98$^{high}$ expressing cells. Ki67$^+$JAK1$^{low}$ cells have lower EBF1 expression (which correlates with high *Ybx3*) compared with the Ki67$^+$JAK1$^{high}$ cells (Supplementary Fig. 4a). When evaluating the expression of Y-box family genes, *Ybx1* was ubiquitously expressed through all developmental stages, with peak expression during the pre-BCRd stage, whereas *Ybx3* expression was selectively induced during the pre-BCRd stage (Supplementary Fig. 3a). We assessed how YBX3 governs B-cell development by using *Ybx3$^{-/-}$* mice with flow cytometry. *Ybx3$^{-/-}$* B-lineage cells had no significant phenotypic defects during the early proliferative phases (Hardy fractions A–C; Supplementary Fig. 4b, d) and late stage differentiation of B cell development (Hardy Fractions D–F; Supplementary Fig. 4c, d). This is consistent with previous findings that YBX1 and YBX3 have redundancies in both function[36] and target mRNA binding[37]. Collectively, our data highlight the reciprocal regulation of *Myc* and *Ebf1* during B-cell development, where MYC is critical for governing differentially expressed genes in the pre-BCRd cluster. Furthermore, pre-BCR signaling limits the IL7R-signaling axis by downregulating *Il7r* expression and JAK1 protein translation.

We next examined the identity and signature of the remaining cycling cells (Fig. 4a). These cells have minimal surrogate light-chain expression (Fig. 2f), indicating further silencing of the locus. Notably, these cells have intermediate expression of the ADT-CD43 (Fig. 4a), which suggests that they are transitioning toward the quiescent CD43$^{lo/-}$ small pre-B-cell stage. Some cells within these clusters also express CD25 protein (Fig. 4a), a marker for pre-B cells[2]. A subset of these cells still expressed high levels of *Nrgn* (Fig. 3a), which may suggest continued pre-BCR signaling. However, unlike the pre-BCR-dependent cluster, these cells have high expression of *Bach2* (Fig. 4b), a transcription factor that restrains antigen-receptor signaling[38,39], and low expression of *Ybx3* and *Myc* (Figs. 3a and 4c). This suggests that these pre-B cells are proliferating independently of pre-BCR signaling. When comparing the differentially expressed genes between the pre-BCR-dependent cells and the pre-BCR-independent cells from cluster I (Pre-BCRi I, Fig. 4a), we found that pre-BCR-independent cells reexpress high levels of *Il7r* and *Ebf1* (Figs. 3b, c, and 4d), in accordance with restrained pre-BCR signaling. When comparing the two different pre-BCR-independent clusters (pre-BCRi I and pre-BCRi II), pre-BCRi I had heightened expression of histone genes (*Hist1h2ae*, *Hist1h2ap*, *Hist1h1b*, *Hist1h1e*, and *Hist1h4d*) (Fig. 4d), but no difference in *Mki67* expression (Supplementary Fig. 3b). On the contrary, the pre-BCRi II cluster is enriched for cells expressing

cell motility-associated actin (*Actg1*), dynein (*Dynll1*), and thymosin (*Tmsb4x* and *Tmsb10*) genes (Fig. 4e). Interestingly, expression of *Cxcr4* is further induced as pre-B cells reach the pre-BCRi stages (Fig. 4b). Expression of CXCR4 would promote targeting of pre-BCRi cells to CXCL12-expressing cells, which also often express high levels of IL7 in the bone marrow[40,41]. Since *Il7r* also increases in pre-BCRi cells, these changes in gene expression suggest that pre-BCRd cells rely on pre-BCR signals for survival/proliferation, while pre-BCRi cells rely on IL7 signals. Although the IL7R and CXCR4 likely promote pre-BCRi cell survival, we did observe that pre-BCRi cells also started to express *Cd74* as well as *Cd44* (Fig. 4e), which complex together to form a receptor for the chemokine ligand, MIF. This could promote chemotaxis of pre-BCRi cells to *Mif*-expressing Fbn1$^{high}$Igf1$^{high}$ osteogenic cells[41,42] that also express IGF1[41], a paracrine growth factor that has been previously shown to be important for the generation of small pre-B cells[43]. Thus, pre-BCRi cells likely require CXCR4/IL7R signals, with a possible contribution from the MIF/CD74/CD44 and IGF1/IGF1R signaling axes, to efficiently transit from a pre-BCR-dependent to a pre-BCR-independent state.

To better characterize the pre-BCR-dependent and pre-BCR-independent populations, we used flow cytometry using the markers identified from our single-cell study. We observed that surface CD74 expression was the highest in the early CD43$^+$ progenitor B cells and had a stepwise decrease in surface expression as the cells matured into mature B cells (Supplementary Fig. 5a). In contrast, intracellular CD74 expression was low in the CD43$^+$ progenitor B cells and small pre-B cells but was the highest in the immature and mature B cells (Supplementary Figure 5a), consistent with transcript expression (Fig. 4b). These results suggest spatiotemporal regulation of CD74 during B-cell development and may point to a differential function of CD74 -serving as a chemokine coreceptor early in development and in antigen processing, as the invariant chain, in more mature stages of development. Finally, to characterize the dynamic *Ebf1* gene expression between the preBCR-dependent and preBCR-independent populations, we used *Myc-GFP* reporter mice to query for EBF1 protein expression. MYC-GFP-expressing progenitor B cells (B220$^+$CD19$^+$CD43$^+$IgM$^-$GFP$^+$) had a heterogeneous EBF1 protein expression (Supplementary Fig. 5b). Compared with the MYC-GFP$^+$EBF1$^{high}$ cells, the MYC-GFP$^+$EBF1$^{low}$ expressing cells also had lower IL7R, CXCR4, and CD74 expression (Supplementary Fig. 5c), which is consistent with transcript expression in the pre-BCR-dependent cluster. To ensure that MYC-GFP-expressing pro-B cells were not the predominant population in the MYC-GFP$^+$EBF1$^{low}$ population, we examined cKIT expression and observed that MYC-GFP$^+$EBF1$^{low}$ cells were primarily cKIT-negative (Supplementary Fig. 5d).

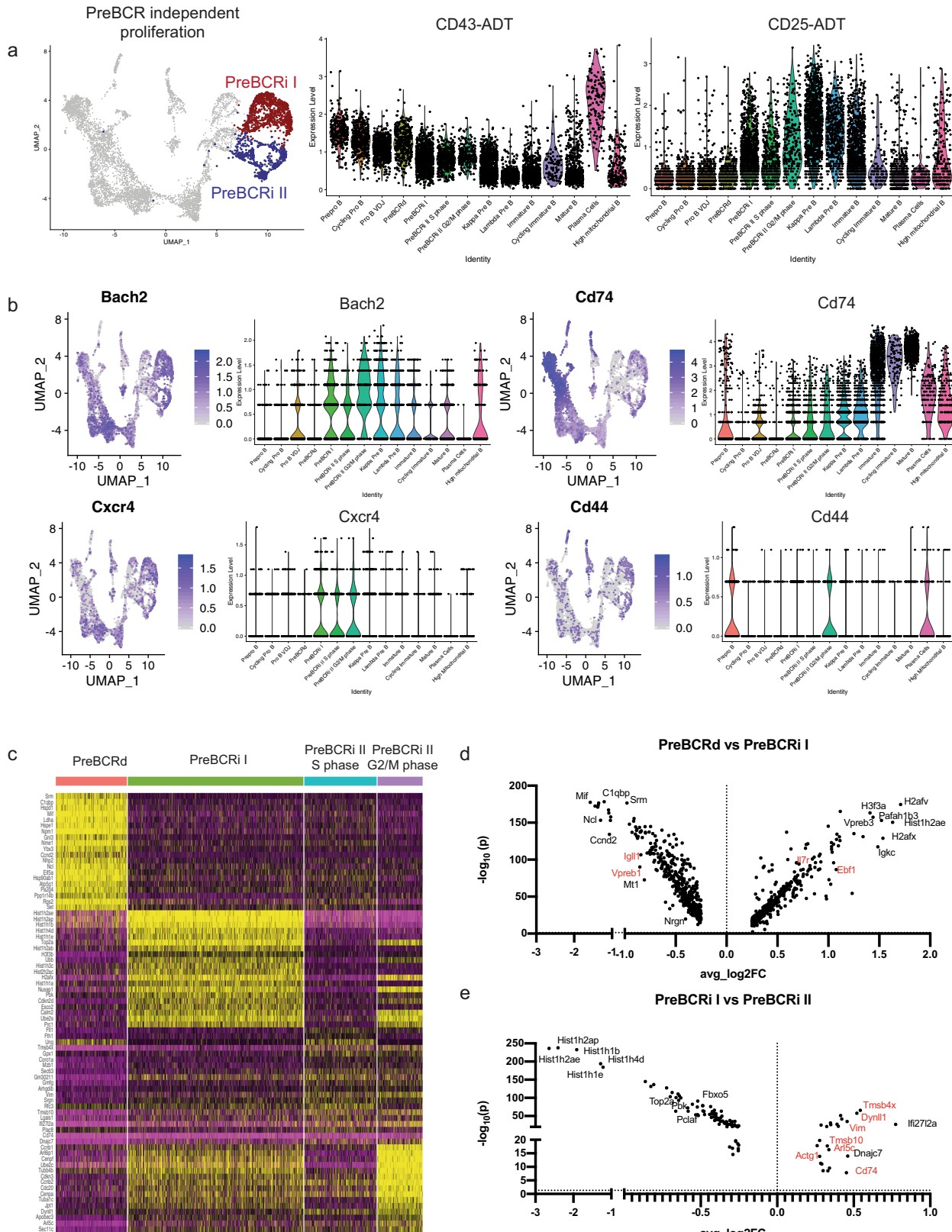

**Fig. 4 Pre-BCR-independent proliferation is distinct from pre-BCR-dependent proliferation. a** Highlighted pre-BCRi I and pre-BCRi II populations (left). Violin plots of expression of ADT-CD43 and ADT-CD25 across the B-cell development stages (right). **b** Feature plot and violin plot for *Cd74*, *Cd44*, and *Bach2*. **c** Heatmap of differentially expressed genes between the pre-BCR-dependent (Pre-BCRd), pre-BCR-independent I (Pre-BCRi I), and pre-BCR-independent II (Pre-BCRi II) S, or G2/M phase. Scale represents normalized counts centered and scaled across cells. **d** Volcano plot showing differentially regulated genes between the pre-BCRd cluster and the pre-BCRi I cluster. **e** Differentially regulated genes between the pre-BCRi I cluster and the pre-BCRi II cluster. Color scales in a, b represent natural log-transformed SCTransform-corrected counts.

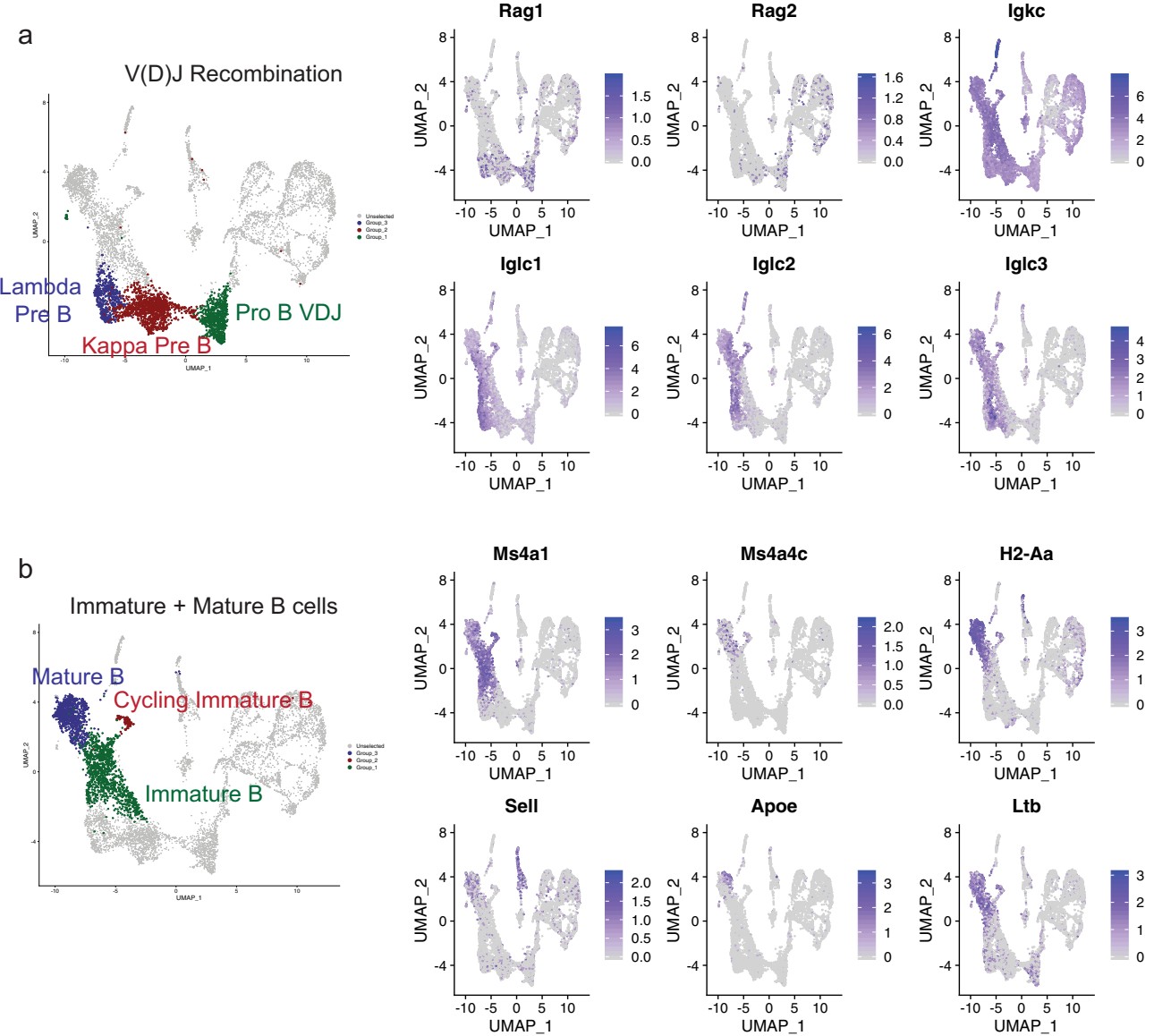

**Fig. 5 B-cell differentiation and maturation. a** Highlighted B-cell clusters undergoing V(D)J recombination (left). Feature plot of genes involved in V(D)J recombination or specific for pre-B-cell expression (right). **b** Highlighted B-cell clusters for late B-cell maturation (left). Feature plot of genes highly expressed in immature or mature B cells (right). Color scale in a,b represents normalized counts centered and scaled across cells.

**B-cell differentiation and maturation.** Quiescent IgM⁻ cells have high expression of *Rag1* and *Rag2*, suggesting V(D)J recombination of pro-B and pre-B cells (Fig. 5a). While *Igkc* expression is detectable in all subsets, except for cycling pro-B cells and pre-BCR-dependent B cells, *Iglc1*, *Iglc2*, and *Iglc3* are only expressed in a subset of cells (Fig. 5a). Furthermore, different B-cell stages exhibited differential expression of trace-element-associated genes such as selenoprotein genes. *Selenom* was expressed in a subset of pro-B VDJ cells (Supplementary Fig. 6a). *Selenop* was expressed in both pro-B and pre-B cells undergoing recombination, whereas *Selenoh* was highly expressed in cycling pro-B and pre-B cells (Supplementary Fig. 6a). The significance of this differential gene expression program remains to be ascertained, although selenium has been associated with immune function and activation[44]. Finally, the IgM⁺ cells were broken down into three clusters corresponding to immature B cells, cycling immature B cells, and mature B cells. The cycling immature B cells have high expression of surface IgM. (Supplementary Fig. 6b). Immature B cells express *Ms4a1* (CD20),

whereas the mature B cells express *Ms4a4c*, *H2-Aa*, *Sell* (L-selectin), and *Ltb* (Fig. 5b). Notably, a subset of mature B cells expressed *Apoe* (Fig. 5b) and showed overlapping detection of both IgM and CD43 (Fig. 1d). This subset shares features with previously described B1 bone marrow B cells[45], although the function of *Apoe* in these cells remains unknown.

**Pseudotime and module analysis of the B-cell development trajectory.** To understand the relationship between developmental stages and changes in gene expression over the B-cell developmental trajectory, we performed pseudotime analysis using Monocle[46]. To lessen the influence of cell cycle on UMAP positioning, we regressed out cell cycle genes within Monocle and performed UMAP dimension reduction. Using stage-defining markers, we identified 13 transcriptionally distinct B-cell developmental stages (Fig. 6a and Supplementary Fig. 7). Compared with the Seurat-based clustering (Fig. 1b), the immature B cell subset was split into two clusters and the preBCRi II S and G2/M

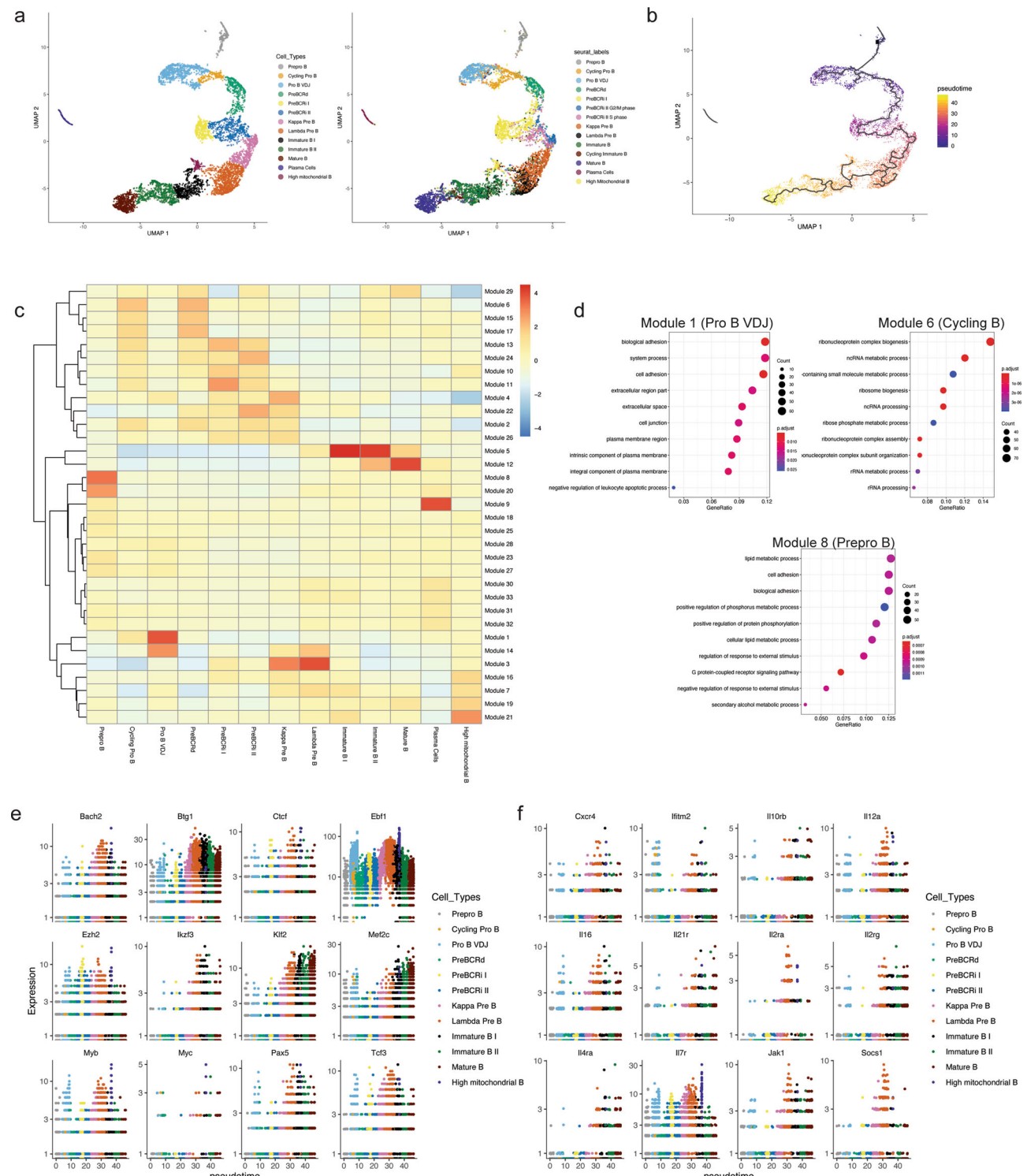

**Fig. 6 Pseudotime analysis illustrates the kinetics of transcription-factor expression and gene modules that are differentially expressed across the B-cell development trajectory. a** Cell cycle-related genes were regressed within Monocle3 and UMAP dimension reduction was performed. Clustering was performed based on the Monocle3 clusters (left). Cells from Monocle-based clustering labeled with Seurat cluster labels (right). **b** Pseudotime values were calculated and plotted. **c** Module analysis to demonstrate gene modules that change across the B-cell developmental trajectory. A total of 33 modules and their expression intensity for each stage are shown. Color scale represents normalized module gene expression. **d** Gene ontology term analysis of selected modules 1, 6, and 8. **e** Expression of B-lymphoid transcription factors and epigenetic factors across B-cell development stages. **f** Expression of cytokine, chemokine and cytokine/chemokine receptors across the B-cell developmental trajectory.

subsets from Seurat were combined into one cluster in Monocle (Fig. 6a). Despite the differing clustering methods and regression of cell cycle genes, the Seurat and Monocle labeling of B-cell subsets demonstrated a high degree of overlap (Fig. 6a and Supplementary Data 3). Pseudotime values were then calculated to establish a developmental trajectory (Fig. 6b). This analysis let us identify modules of genes that are changing across the developmental trajectory (Fig. 6c). Gene ontology analysis of these gene modules suggested that the early B-cell stages, including pre–pro B and pro-B cells, are significantly enriched for cell-adhesion processes, whereas these signals are diminished in pre-B cells (Module 1 and Module 8, Fig. 6c, d). This is in accordance with previous findings that pro-B cells strongly promote adhesion to IL7-producing stromal cells[47], whereas pre-BCR-signaling and its downstream target IKZF1, are important for downregulating stromal adhesion components[7,48]. Furthermore, the cycling pro-B cells and pre-BCR-dependent cells were enriched for genes involved in metabolic processes (Module 6; Fig. 6d), suggesting metabolic reprogramming after major checkpoints (B-cell lineage commitment and pre-BCR selection, respectively) during B-cell development. Finally, evaluation of the expression of transcription factors, epigenetic modifiers (Fig. 6e), cytokines, chemokines and their respective receptors (Fig. 6f) indicates differential expression kinetics across the B-cell development trajectory. Thus, our findings confirm some previous observations, but also identify distinct gene programs that exhibit highly dynamic changes during B-cell development.

**Stage-specific gene expression networks correlate with various human B-ALL subtypes and prognosis.** Defects in B-cell differentiation and dysregulation of signaling pathways lead to B-cell transformation. To identify B cell developmental pathways that are associated with human B-ALL subtypes, we used differentially upregulated gene markers identified in both the Seurat and Monocle analyses. Using averaged z-scores of upregulated genes from each cluster, the transcriptional network of each stage was compared with various human B-ALL subtypes. This resulted in the hierarchical clustering of proliferation and differentiation stages (Fig. 7a).

Interestingly, the human B-ALL subtypes that were enriched for proliferating pro-B, pre-BCRd, and preBCRi clusters, included the BCL2/MYC, IKZF1 N159Y, and KMT2A subtypes, which are all associated with high risk poor prognosis[49–51] (Fig. 7a). In contrast, the B-ALL subtypes that were enriched for genes characteristic of other B cell differentiation stages, such as ETV6–RUNX1 and ZNF384 rearrangements, were the ones with favorable outcomes[52–55] (Fig. 7a). Performing hierarchical clustering using only the Monocle gene list gave identical clustering patterns. We then performed statistical testing for significance of the hierarchical clustering[56]. Testing of the row-based hierarchical clustering of different developing B-cell stages showed statistically significant differences in the proliferating B cells versus the differentiation stages (Fig. 7b, left, $P = 0.006$, Monte Carlo test corrected for familywise error rate). Notably, the preBCRd stage was also distinct from the preBCRi and proliferating pro-B stage (Fig. 7b, left, $P = 0.006$, Monte Carlo test corrected for family-wise error rate). Likewise, column-based hierarchical clustering testing of the human B-ALL subtypes demonstrated that the high-risk leukemias that correlated with proliferative B cells were significantly distinct from the low-risk leukemias that correlated with differentiating B cells (Fig. 7b; right, $P = 0.001$; Monte Carlos test corrected for family-wise error rate). The pre-BCR module genes, including YBX3 and NRGN, are significantly upregulated in BCL2/MYC, IKZF1 N159Y, and MEF2D B-ALL subtypes (Fig. 7c). Despite the numerous

pre-BCR module genes highly expressed in various B-ALL subtypes, not all pre-BCR genes share this pattern, indicating transcriptional heterogeneity of B-ALLs compared with normal-developing B cells (Fig. 7c, Supplementary Fig. 8a). Finally, we examined whether YBX3 expression correlated with outcome in B-ALL. We observed that pediatric B-ALLs with above-median YBX3 expression are associated with worse prognosis (Fig. 7d, hazard ratio = 2.03, $P = 0.032$, log-rank test), whereas adult B-ALL patients (that mainly comprise Ph+ B-ALLs) show no significant difference in survival (Fig. 7d). Overall, we identify B-cell gene expression networks that are modulated during B-cell development (Fig. 7e) and correlate with human B-ALL subtypes. In addition, we specifically demonstrate that a YBX3-related module is associated with a poor prognosis in human B-ALLs.

**Discussion**

The evaluation of different organs[57] and niches[41,58] at single-cell resolution has greatly expanded our understanding of the cellular diversity that is present within the bone marrow. However, these previous studies have not resulted in a detailed description of B-cell development. This is due to the paucity of developing B cells in these broad surveys of the bone marrow compartment. B-cell development has also been examined using cell surface markers in conjunction with flow cytometry[1]. These studies, in conjunction with in vivo reconstitution experiments to ascertain precursor–progeny relationships, have provided a general outline of B-cell development[1,59]. Furthermore, resources such as Immgen provide a wealth of transcriptional data about various sorted B-cell compartments[60]. However, these sorted-cell populations are relatively heterogeneous and thus fail to provide detailed single-cell resolution of B-cell development. In this study, we coupled the conventional surface marker-based staging of B-cell development using CITE-Seq with single-cell transcriptomics to identify unappreciated transcriptional heterogeneity during B cell development and link them to various underlying biological processes. In addition, we identify the RNA-binding protein YBX3 as a marker of pre-B cell differentiation and correlate Ybx3 expression with outcomes in patients with B cell acute lymphoblastic leukemia.

Proliferation and differentiation during B-cell development are important biological modalities that underpin selection and BCR-repertoire diversification. Given our sorting strategy (Supplementary Fig. 1a), we did not capture all possible subsets of B cells, such as B220−CD19+ B1 progenitor cells or CD11c-expressing B cells. Nonetheless, our results highlight the diversity of transcriptional networks that are present within both proliferation and differentiation modalities during B-cell development. Specifically, the proliferating B cells were clustered tightly and labeled as G1PM, S, and G2/M-phase cells (Fig. 1c) based on their transcript status using a predetermined cell cycle gene list. The postmitotic G1 phase (G1PM) was recently described in a study demonstrating that stimulated splenic B cells undergo mitogen-independent proliferation in which genes associated with G2/M phase are not fully extinguished during the G1 phase[17]. Thus, proliferating B cells can undergo an extremely short G1 stage that bears features of G2/M. Our study highlights that proliferating progenitor B cells also exhibit a G1 phase containing many G2/M-signature genes (Fig. 1c and Supplementary Fig. 2a), which is consistent with rapidly proliferating B cells with very short G1 stages.

Our studies demonstrate unexpected dynamic changes in transcription factors that orchestrate B-cell development. We were largely able to confirm the expression kinetics for the critical B-lymphoid lineage transcription factors Pax5 and Ikzf1 across B cell development stages (Fig. 3b). In contrast, Ebf1, another key

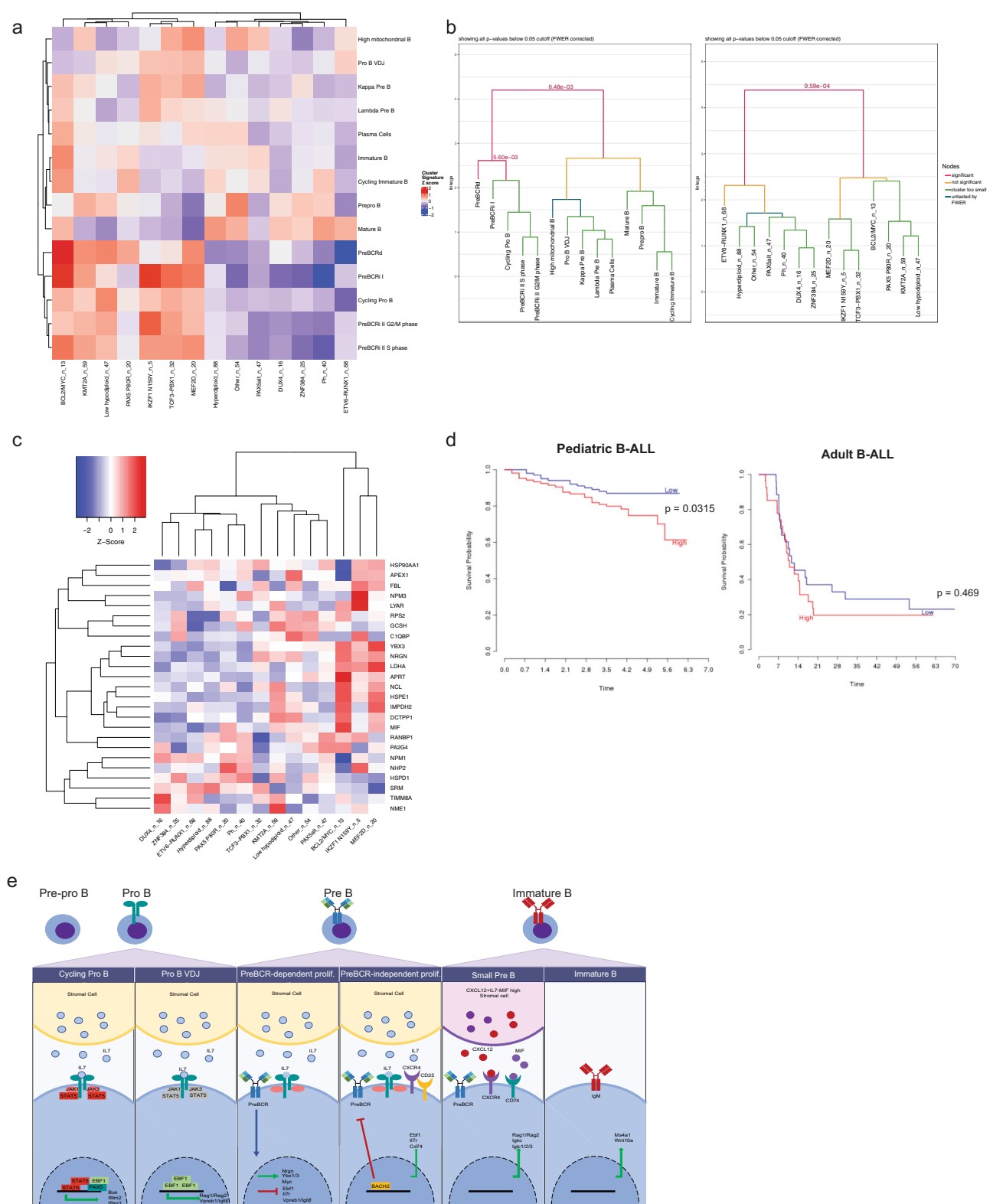

B-cell transcription factor, has a strikingly large dynamic range of expression (Fig. 3b) that varies greatly over the course of B-cell development. *Ebf1* has the highest expression during both heavy and light chain recombination stages, intermediate expression during cytokine-mediated proliferative stages, and the lowest expression during pre-BCR and BCR-signaling stages (Fig. 3b). Likewise, the expression of EBF1-target genes exhibited the same patterns (Fig. 3c). This points to dynamic changes in *Ebf1* in

regulating key gene expression networks throughout B-cell development. Furthermore, B-lymphoid transcription factors, including *Ebf1*, have been shown to serve as metabolic gate-keepers, where they repress genes encoding proteins for glucose uptake and utilization, and thereby prevent malignant transformation[61]. In light of this, we provide evidence that EBF1 represses *Myc* and that low gene expression of *Ebf1* correlates strongly with the activation of pre-BCR-dependent gene modules

**Fig. 7 B-cell developmental gene networks correlate with various B-ALL subtypes and associate with prognosis. a** Heatmap of cluster gene sets and various human B-ALL subtypes. n_number denotes the number of patient samples falling in each B-ALL subtype. Average z-score of cluster marker genes was calculated for each cluster and hierarchical clustering was performed for both B-ALL subtype and average cluster z-score. **b** Monte-Carlo-based statistical testing of the hierarchical clustering was performed for row (B-cell development stages) and column (human B-ALL subtypes). Significant *p*-values (<0.05) are shown and colored as a red line. **c** Overlapping differentially upregulated gene markers in the pre-BCR-dependent clusters from Seurat and Monocle are plotted (y axis). The expression of these genes was queried across various B-ALL subtypes (x axis). The number of B-ALL samples (*n*) is listed after each subtype. **d** Survival curve of pediatric B-ALL (left, time in years) and adult B-ALL (right, time in months). Low and high expression correspond to below-median and above-median expression of YBX3, respectively. Survival data were obtained from Prediction of Clinical Outcomes from Genomic Profiles (PRECOG)[71] **e** Proposed model of B-cell development.

that promote metabolic reprogramming (such as amino acid transporters) and proliferation. Therefore, the expression kinetics of *Ebf1* are tightly controlled to enable the unique aspects of alternating cycles of proliferation and differentiation throughout B-cell development.

Our single-cell RNA-Seq data pointed to an unappreciated expression of the RNA binding protein YBX3 in pre-B-cell differentiation and proliferation. Notably, *Ybx3* expression was significantly upregulated during the pre-BCR dependent stage (Fig. 3a). Likewise, the related YBX family member *Ybx1* is also most highly expressed at the pre-BCRd stage (Fig. 3a). Analysis of *Ybx3*$^{-/-}$ mice did not show major developmental defects (Supplementary Fig. 4b, c) and thus YBX1 likely serves a redundant function with YBX3 in developing B cells. Therefore, YBX1 and YBX3 may initiate the MYC-dependent transcriptional program that characterizes the pre-BCR-dependent expansion stage[29].

Elucidating the requirements and regulation of normal B-cell development has significantly improved our understanding of how developing B cells can undergo transformation and maintain leukemic states. B-ALL subtypes exhibit significant transcriptional diversity[12]. While differences between mouse and human B cell populations exist, many aspects of mouse and human B cell development are highly analogous. Our comparative analysis between stage-defining gene expression in wild-type progenitor B cells and different human B-ALL subtypes suggests that cycling pro-B, pre-BCR-dependent, and preBCR-independent subsets shared the highest similarities with high-risk human B-ALL subsets, including BCL2/MYC, IKZF1 N159Y, and KMT2A-rearranged leukemias. In contrast, the absence of these cycling-stage signatures was associated with low-risk human B-ALL subsets, including ETV6-RUNX1 and ZNF3840-rearranged leukemias (Fig. 7a). Importantly, while these cycling stage gene signatures do include some canonical cell cycle genes, they are also characterized by non-cell cycle related unique gene sets associated with these specific developmental stages, which may promote a more aggressive disease state (Supplementary Fig. 8). Therefore, our findings suggest that the presence or absence of stage-specific genes may associate with prognosis and could be targeted to further improve outcome.

When looking at individual genes from each cluster in different B-ALL subtypes, we found both overlapping and nonoverlapping gene modules, suggesting that B-cell leukemias partially associate with specific gene expression modules of multiple B-cell development stages. In particular, we found that pre-BCR module genes, including *Nrgn* and *Ybx3*, were significantly upregulated in the high-risk BCL2/MYC, IKZF1 N159Y, and MEF2D B-ALL subtypes (Fig. 7b). In addition to our findings on the importance of MYC during pre-BCR signaling, previous studies have demonstrated the upregulation of IKZF1 and MEF2D[62] upon pre-BCR signaling. Therefore, aberrant dysregulation through mutations and rearrangements in *IKZF1* and *MEF2D*, respectively, may lead to the heightened usage of pre-BCR-signaling related genes for transformation. Importantly, we were able to demonstrate that high expression of *Ybx3*, a key component of

the pre-BCR-signaling module, correlates with poor prognosis in pediatric B-ALL (Fig. 7c). Interestingly, high expression of *Ybx3* also correlates with dramatic downregulation of *Ebf1*. The combination of elevated YBX3 and reduced *EBF1*[63] expression likely establishes the MYC-driven transcriptional program that contributes to pre-B-cell transformation. As YBX3 has been shown to repress JAK1 translation[28], this may explain the requirement for ectopic activation of the JAK/STAT5 pathway to overcome a potential YBX3-driven negative feedback loop. Collectively, our data and previously published findings converge on multiple mechanisms that can activate different proliferation modules, leading to high risk-leukemia and a poor prognosis. Using strategies that target YBX3 or pre-BCR-related modules in pre-BCRd-related leukemias may also further improve outcomes in patients with these subsets of high-risk B-ALL.

## Methods

**Animals**. All animals used were bred and housed in specific pathogen-free facilities at the University of Minnesota and animal experiment protocols were approved by Institutional Animal Care and Use Committees (IACUC 2010-38515A and IACUC 1904-36975A). All of the animals used were 6- to 12-week-old C57BL/6 J males and females with appropriate age- and sex-matched controls. Specifically, the two wild-type mice used for the single-cell RNA-seq experiment were C57BL/6 J males, CD45.2, and 8-week-old mice. The mice used for assessing B cells that were potentially excluded from analysis due to flow sorting gating scheme, were C57BL/6 J males, CD45.2, that were 7–10-weeks old. The JAK1 and EBF1 flow experiment was performed with C57BL/6 J male and female mice, CD45.1 and CD45.2, that were 7-weeks old. The *Ybx3*$^{-/-}$ phenotyping experiments were performed with C57BL/6 J male and female mice, CD45.2, and 6–12-weeks old. *Myc*-GFP mouse experiments were performed with C57/BL6 male and female mice, CD45.2, that were 8–10-weeks old. The *Ybx3*$^{-/-}$ mice were graciously provided by Dr. Timothy Ley at Washington University in St. Louis and have been previously described[36]. *Myc-GFP* reporter mice were obtained from The Jackson Laboratory. All animals were housed in a dark/light cycle of 14 h/10 h. Light cycle was from 6 AM to 8 PM. Ambient temperature was at 72Fᵒ with a humidity ranging from 30 to 40%.

**Tissue processing and cell preparation**. For flow-cytometry and cell-sorting experiments, bilateral femurs and tibias were harvested from mice. Bones were flushed with 1X PBS with 2% fetal bovine serum (Sigma Aldrich, 12133 C), 0.1% sodium azide (Ricca, 7144.8-16), and 0.5 mM ethylenediaminetetraacetic acids (EDTA, Fisher Scientific, S3113), pH 7.4. The cells were filtered through a 70 μm mesh, centrifuged at $350 \times g$ for 5 min, and then incubated for 5 min with 5 mL of ACK lysis buffer (0.15 M ammonium chloride (Fisher, A661), 10 mM potassium bicarbonate (Fisher, P184), 1 mM EDTA (Fisher Scientific, S3113)). for red-blood cell lysis. Cells were then washed and centrifuged at $350 \times g$ and subsequently resuspended for cell counting on a hemocytometer (Fisher Scientific. 02-671-10) and staining.

**Flow cytometry and antibodies**. All flow cytometry was performed using the BD Fortessa cytometers (BD Biosciences) in the University of Minnesota Flow Cytometry Core. Bone marrow cells obtained using the method above were stained with various FACS antibodies, including B220-BUV395 (RA3-6B2, BDBiosciences, 563793, 1:100), B220-Pacific Blue (RA3-6B2, BDBiosciences, 558108, 1:100) B220-PE-Cy7 (RA3-6B2, BioLegend, 103222, 1:100), CD11c-APCef780 (N418, ThermoFisher, 47-0114-82, 1:100), CD11c-BUV395, (N418, BD Biosciences, 744180, 1:100), GhostRed780 (Tonbo Biosciences, 13-0865, 1:1000), Ter119-APCef780 (TER-119, ThermoFisher, 47-5921-82, 1:100), NK1.1-APCef780 (PK136, ThermoFisher, 47-5941-82, 1:100), NK1.1-PE (PK136, eBioscience, 12-5941-82, 1:100), Ly6G-APCef780 (RB6-8C5, ThermoFisher, 47-5931-82, 1:100), Ly6G-BV421 (RB6-8C5, BioLegend, 127627, 1:100), CD4-APCef780 (GK1.5, ThermoFisher, 47-0041-82, 1:100), CD8-APCef780 (53-6.7, ThermoFisher, 47-0081-82, 1:100),

CD43-Biotin (S7, BDBiosciences, 553269, 1:100), CD43-BV786 (S7, BDBiosciences, 740857, 1:100), CD19-BV605 (6D5, BioLegend, 115540, 1:100), IL7R-BV421 (A7R34, BioLegend, 135023, 1:100), CXCR4-PerCP-Cy5.5 (L276F12, BioLegend, 146509, 1:100), CD74-BUV395 (In-1, BDBiosciences, 740274, 1:100), CD98-PE-Cy7 (RL388, BioLegend, 128124, 1:100), and cKIT-PE-Cy7 (2B8, eBioscience, 25-1171-82, 1:100). In short, surface staining was performed for 20 minutes with FACS antibodies on ice, washed and either analyzed or stained for intracellular FACS antibodies. For intracellular staining of JAK1-AF488 (413104, R&D Systems, IC4260G, 1:100), EBF1-PE (T26-818, BD Biosciences, 565494, 1:100), and anti-GFP-FITC (Rockland, 600-402-215, 1:400) surface stained cells were fixed/permeabilized using the eBioscience Transcription Factor staining kit (eBioscience, 00-5523-00) for 30 minutes at room temperature, washed, and then stained for 30 minutes in permeabilization buffer. Cells were then washed and resuspended in 1X PBS with 2% fetal bovine serum (FBS), 0.1% sodium azide and 0.5 mM ethylenediaminetetraacetic acid, pH 7.4, for flow cytometric analysis. Cell sorting was performed on a BD FACSAria sorter (BD Biosciences). All flow-cytometry data acquired were analyzed using FlowJo software (Tree Star)

**Cell hashing and CITE-Seq.** Bone marrow cells from two 8-week-old wild-type mice were each stained with 1 µg of different hashtag antibodies, TotalSeq A0301 (M1/42, BioLegend, 115801, 1 µg) and TotalSeqA0302, (M1/42, BioLegend, 115803, 1 µg). At the same time, cells were stained with FACS antibodies: B220-Pacific Blue (RA3-6B2, BDBiosciences, 558108, 1:100), CD43-Biotin (S7, BD Biosciences, 553269, 1:100), CD11c-APCef780 (N418, ThermoFisher, 47-0114-82), Ter119-APCef780 (TER-119, ThermoFisher, 47-5921-82, 1:100), NK1.1-APCef780 (PK136, ThermoFisher, 47-5941-82, 1:100), Ly6G-APCef780 (RB6-8C5, Thermo-Fisher, 47-5931-82, 1:100), CD4-APCef780 (GK1.5, ThermoFisher, 47-0041-82, 1:100), CD8-APCef780 (53-6.7, ThermoFisher, 47-0081-82, 1:100), GhostDye Red780 viability dye (Tonbo Biosciences, 13-0865, 1:1000) and 1 µg of CITE-Seq antibodies: B220 (TotalSeq-A0103, RA3-6B2, BioLegend, 103263), CD19 (TotalSeq A0093, 6D5, BioLegend, 115559), CD93 (TotalSeq A0113, AA4.1, BioLegend, 136513), CD25 (TotalSeq A0097, PC61, BioLegend, 102055) and IgM (TotalSeq A0450, RMM-1, BioLegend, 406535). Cells were stained with the above-mentioned antibodies for 20 minutes on ice, washed, and resuspended in 1X PBS with 2% fetal bovine serum (FBS), 2 mM ethylenediaminetetraacetic acid, and pH 7.4 buffer containing 1 µg of streptavidin-PE (TotalSeq A0113, BioLegend, 405251), which served the dual purpose of cell sorting and CITE-Seq for the CD43 antigen expression. scRNAseq was performed in parallel to FACS analysis.

**Single-cell RNA sequencing.** For 10X Genomics scRNA-seq, we generated three libraries that measure (1) mRNA transcript expression (RNA), (2) mouse-specific hashtag oligos (HTO), and (3) cell surface marker levels using antibody-derived tags (ADT). Cells were harvested and stained as described above. In order to have balanced populations of various B-cell development stages, we enriched for earlier progenitor B cells (B220$^+$CD43$^+$) by sorting at a 1:1 ratio of Dump$^-$B220$^+$CD43$^+$ cells and Dump$^-$B220$^+$CD43$^-$ cells. A total of 20,000 cells per mouse were sorted into a single microtube containing 1X PBS with 50% FBS and were washed and resuspended in 1X PBS with 10% FBS prior to cell capture. The sample was split into three libraries (RNA, HTO, and ADT). Reverse transcription PCR and library preparation were carried out under the Chromium Single Cell 3′ v3 protocol (10X Genomics) as per the manufacturer's recommendations. After library prep, quality control was performed using a bioanalyzer (Agilent 2100 Bioanalyzer, Agilent Technologies) and preliminary sequencing of the RNA library on a MiSeq (Illumina) to determine the approximate number of cells and general quality. After passing quality control, the library was sequenced on the NovaSeq 6000 with 2 × 150-bp paired-end reads (Illumina). Raw and processed data have been deposited at Gene Expression Omnibus and are available via GEO accession GSE168158. The code used in this study can be obtained upon request.

**Single-cell bioinformatic analyses.** Raw sequencing data were processed using the CellRanger pipeline (version 3.1.0, 10X Genomics) "mkfastq" to demultiplex the three Illumina libraries (RNA, HTO, and ADT) and "count" was used to align reads to the mouse genome (mm10, provided by 10X Genomics, ver 3.0.0) and generate mRNA transcript, HTO, and ADT count tables. Raw count data were loaded into R (v. 3.6.1) and analyzed with the Seurat R package (v 3.0.3.9039). The RNA dataset was filtered to include only GEMs (gel beads in emulsion, which are oil droplets containing uniquely barcoded beads that ideally contain one individual cell) expressing more than 300 genes (counts > 0) and genes expressed in more than 3 GEMs (counts > 0). The proportion of mitochondrial RNA in each GEM was calculated and GEMs with extreme levels (top 0.5% of all GEMs) were removed from the analysis. For the remaining GEMs, the HTO count table was added to the dataset and normalized by a centered-log ratio method. Multiplets (i.e., GEMs containing one or more cells) were discovered using two independent orthogonal methods: (1) DoubletFinder software (version 2.0.3)[64] utilized only the RNA expression dataset to predict GEMs as doublets or singlets and (2) the HTO sample-based expression dataset was supplied to the HTODemux Seurat function for doublet classification. The default DoubletFinder analysis method searched for optimal classification parameters and returned 8,209 singlets (3458 WT-1 or 3103 WT-2) and 4427 doublets. The HTODemux function clustered GEMs by their

HTO expression levels, resulting in six major clusters of singlet GEMs representing each of the individual HTOs in the experiment (i.e., two HTOs represented WT-1,2 and four other HTOs were present, but excluded to derive only the wildtype B cells). This function was tested using a range of initial k-values (7–26), where k = 22 provided the cleanest classification results. Four other hashtag samples were present in this experiment and were excluded to derive only the wildtype B cells. A total of 7454 GEMs contained WT singlets (3902 WT-1 or 3552 WT-2) and 5182 GEMs contained HTO multiplets. Cross table comparison between DoubletFinder and HTODemux results demonstrated 893 WT GEMs called doublets by DoubletFinder and singlets by HTODemux. Conversely, 1576 WT GEMs were labeled doublets by HTODemux and singlets by DoubletFinder. This latter discrepancy along with the difficulty of DoubletFinder to accurately identify transcriptionally similar doublets led us to exclude the DoubletFinder classifications going forward. Using only the HTODemux classifications, GEMs identified as multiplets or negative were removed from further analysis. The WT singlets expressed a median of 1409 genes with a median of 3548 counts. For the WT singlets, the raw RNA counts were transformed using the Seurat function "SCTransform"[65] including the percent of mitochondria expression as a regression factor. Principal components analysis (PCA) was performed using the normalized SCT dataset (RunPCA function) and two-dimensional representations were generated using the top 30 PCA vectors as input to the RunTSNE and RunUMAP functions. Cells were clustered using the FindNeighbors function (top 30 PCA vectors) and FindClusters function (testing a range of possible resolutions: 0.05, 0.1, 0.15, 0.2, 0.25, 0.3, and 0.4). A final resolution of 0.4 was used to classify cells into gene expression clusters. The robustness of cluster classifications was validated using the significant DE lists. Any pairwise cluster comparison that had fewer than 5 differentially expressed genes was merged into a newly labeled joint cluster, and the process of pairwise DE testing would begin again using the revised cluster classifications. At the final cluster resolution of 0.4, there were zero cluster merging steps. Each cell was classified according to its expression of canonical cell cycle genes using the Cell-CycleScoring function. The CellCycleScoring function was used to calculate the average expression of three different cell cycle phase related gene sets: S, G2M (genesets provided in Seurat), and a recently described G1 postmitotic (G1PM) gene set (Birc5, Myc, Mki67, Foxm1, Aurkb, and Plk1)[17]. Cell cycle scores were calculated from the SCT normalized expression values and used as input to discretely classify each cell by phase. The highest cell cycle score was used to label each cell by phase, unless all three scores were negative, resulting in the default G0/G1 classification. These S and G2M phase scores for each cell were used as additive factors in a linear model of gene expression (i.e., when regressing out the influence of cell cycle).

Single cell surface protein expression data (ADT) were filtered to include only the WT singlets and counts were normalized according to the centered-log-ratio method in Seurat. For each of the six markers measured (B220, CD19, CD93, CD25, IgM, and CD43), the normalized counts were centered (subtracting the mean expression from each value) and scaled (dividing centered value by standard deviation) across all cells. The Seurat object with S and G2M phase scores and a resolution = 0.4 was converted into a cell_data_set object for use with the Monocle (v3) R package[66]. The aligncds function was used with residual_model_formula_str = "~S.Score + G2M.Score" to adjust for the cell cycle status. After adjustment for cell cycle status, UMAP dimensional reduction and clustering were performed in Monocle. The final resolution for Monocle clustering was 0.0009. This resolution resulted in 3 separate partitions for the clusters. The plasma cell cluster was in its own partition and was excluded from the Pseudotime trajectory analysis. The remaining two partitions were (1) pre–pro B cells and (2) all other cell populations. We relabeled the partitions so that the pre–pro B cells and the remaining cell populations of interest would be in the same partition for pseudotime analysis. We initiated the pseudotime trajectory in the pre–pro B cell population, using a custom function described in the Monocle 3 documentation (get_earliest_principal_node) to automatically select the starting point for the pseudotime analysis. To compare the Monocle and Seurat clustering results, the number of cells and frequency of cluster membership overlap between Monocle and Seurat was calculated.

We used Monocle 3 methods to identify modules of coregulated genes. First, we used the function graph_test to identify variable genes in the data using the Moran's I statistic. Then we identified the genes that had a significant q-value (<0.05) from the autocorrelation analysis and then grouped these genes into modules using UMAP and Louvain community analysis. We used the enrichGO function in the clusterProfiler package (v 3.14.3) to evaluate enrichment of the modules in GO terms across all three ontologies (BP, CC, and MF)[67,68].

**Landscape in silico deletion analysis (LISA) and gene set enrichment analysis (GSEA).** The top 100 differentially upregulated genes obtained from Seurat FindAllMarkers function were used as input into the LISA Cistrome (lisa.cistrome.org). The transcription factor ChIP-Seq dataset was used to infer the transcriptional regulators for differentially regulated genes of the pre-BCR-dependent proliferation stage. We used the Gene set enrichment analysis (GSEA) software to identify gene sets from the Molecular Signature Databases[69,70], comparing the differentially expressed genes in the pre-BCR-independent proliferating cells versus the pre-BCR-dependent cells. The MSigDB (v 7.0) annotations were extracted from the msigdbr R package (v 7.0.1).

GSEA and gene ontology analysis was completed using the "enricher" function from the clusterProfiler R package (v 3.14.0).

**Enrichment analysis of single cell RNA-Seq cluster marker genes in human B-ALL gene expression.** We used enrichment analysis to examine whether human B-ALL subtypes have similarity to particular B cell progenitor populations. We created robust signatures for each cell type using the Seurat and Monocle top cluster marker genes independently, and we also created combined cluster marker gene sets by taking the intersection of the independent Seurat and Monocle cluster gene signatures for each of the different cell types we identified in the single cell data. The top marker genes are the genes that are differentially expressed for a given cluster when compared with all other clusters. Default approaches for finding top marker genes were used for both Seurat and Monocle. The findAllMarkers() function from Seurat was used as described in the previous section, "*Single-cell Bioinformatic Analyses*", to identify which genes are differentially expressed in one cluster compared with all other clusters for the Seurat cluster. The top_markers() function from Monocle was used to identify which genes are differentially expressed in one cluster compared with all others for the Monocle cluster definitions. We used the Monocle default parameters, so 25 genes were returned per Monocle cluster with the top_markers() function. We did not apply any additional filters to the top gene markers for the Monocle analysis. For the Seurat cluster marker genes we only used genes with positive log fold changes to better match the Monocle cluster gene marker results. We did not include downregulated marker genes from the Seurat analysis. For both Monocle and Seurat cluster gene lists we had to convert the mouse gene names to human gene names, so some genes were removed that did not have appropriate human homologs. Additionally, when there was not a direct 1:1 mapping between the Monocle and Seurat clusters, cluster gene lists from one method were merged before comparison to the other method's cluster gene list. We then used the intersection of the Monocle lists and the Seurat lists to get our final list of genes for each cluster for our individual B-ALL heatmaps for each group of cluster-marker genes.

We downloaded publicly available B-ALL bulk RNA-Seq count data[12]. We summed expression values for each gene across biological replicates for each B-ALL subtype to create an average sample for each sub-type. We normalized and log2-transformed the data to create log2cpm values for unsupervised hierarchical clustering of each combined cluster marker gene set against all B-ALL subtypes. To gain a bigger picture view of the association between gene networks and human B-ALL subtypes, we created a single heatmap showing a summarized cluster gene set z-score for each B-ALL subtype. We calculated each cluster gene set z-score by averaging the z-scores of the cluster marker genes in each individual cluster gene set for each B-ALL subtype. In the heatmap of these data we allowed for both column (B-ALL subtype) and row (average cluster z-score) hierarchical clustering.

**Statistical analysis.** Differential gene expression (DE) analysis was completed using the FindMarkers function, employing a Wilcoxon rank-sum test between all pairwise clusters or between a single cluster vs. all others. Genes were considered significant if the absolute value of log2-fold-change was $> = 0.25$ and Bonferroni-adjusted $p$-value $< = 0.01$. The statistical significance of hierarchical clustering for the human B-ALL heatmaps was done using a Monte-Carlo based method[56]. Specifically, the sigclust2 package was used with the parameters of Euclidean distance and complete linkage for clustering methods. Data and statistical analyses were performed using Prism 8 (Graphpad). A Shapiro-Wilk test was performed to assess data normality, and unpaired data that passed normality were analyzed using an unpaired student $t$-test. Paired data derived from the same mouse were analyzed using a parametric paired student $t$-test.

**Reporting summary.** Further information on research design is available in the Nature Research Reporting Summary linked to this article.

## Data availability
The data supporting the findings of this study are available within the paper and supplementary information file. Single-cell RNA-Seq data were deposited at Gene Expression Omnibus, with the primary accession code: GSE168158. Mouse genome mm10 was used as reference sequence (https://www.ncbi.nlm.nih.gov/assembly/GCF_000001635.20/). EBF1 ChIP-Seq data for Fig. 3e were obtained from GSM2863146. Bulk RNA-seq data for wild type and Pax5$^{+/−}$ x Ebf1$^{+/−}$ leukemia for Fig. 3f was obtained from GSE148680. Source data are provided with this paper.

## Code availability
No custom code was used or generated in this paper, as publicly available software and code was used (Seurat, Monocle3, DoubletFinder, GSEA, and GO-term analysis). Code can be provided upon request.

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

## Acknowledgements

We thank G. Hubbard, A. Rost, and N. Keller, for mouse and technical assistance, J. Motl and P. Champoux for cell sorting and Flow Cytometry Core Facility maintenance at the University of Minnesota (5P01AI035296), D. George for providing *Ybx3*$^{-/-}$ bone marrow, E. Stanley, J. Daniels, and Dr. K. Beckman and the University of Minnesota Genomics Center for 10X genomics single-cell capture and sequencing, Dr. Meinrad Busslinger for *Pax5*$^{+/-}$ mice, Dr. Rudolf Grosschedl for *Ebf1*$^{+/-}$ mice, Dr. Tim Ley for *Ybx3*$^{-/-}$ mice, and Drs. J. Pereira, T. Lebien, and D. Owen for review and comments on the paper. The authors acknowledge the Minnesota Supercomputing Institute (MSI) at the University of Minnesota for providing resources that contributed to the research results reported within this paper. This work was supported by an individual predoctoral F30 fellowship from the NIH (F30CA232399) and T32 training grant (T32 GM008244), R.D.L, and. NIH grants R01AI124512, R01AI147540, and R01CA232317, M.A.F.

## Author contributions

Conceptualization, R.D.L. and M.A.F.; Methodology, R.D.L., and M.A.F.; Formal Analysis, R.D.L., S.A.M., T.P.K., R.S.L., L.H.H., and M.A.F.; Investigation, R.D.L, L.H.H.; Resources, M.A.F.; Data Curation, S.A.M., T.P.K., and R.S.L.; Writing—Original Draft, R.D.L. and M.A.F.; Writing—Review & Editing, R.D.L, S.A.M., T.P.K., R.S.L., M.A.F.; Visualization, R.D.L. and L.H.H.; Supervision, M.A.F.; Project Administration, M.A.F.; Funding Acquisition, R.D.L. and M.A.F.

## Competing interests

The authors declare no competing interests.
