## [Peer Review File · Nature Communications]

Single-cell analysis of developing B cells reveals dynamic gene expression networks that govern B cell development and transformationREVIEWER COMMENTS

Reviewer #1 (Remarks to the Author):

This study used single-cell transcriptomics and proteomics to characterize B cell development. The analysis revealed unique transcriptional signatures that characterize the pre-B cell expansion stages into novel pre-BCR-dependent and pre-BCR-independent phases. These changes correlate with unexpected dynamic changes in expression of the transcription factor EBF1 and the RNA binding protein YBX3, that are defining features of the pre-BCR-dependent stage. Detailed analyses also characterize the expression kinetics across B cell development of transcription factors, cytokines, chemokines, and their associated receptors. An important set of findings in this Ms point to key developmental nodes linked to B cell transformation (ALL).

the paper is noteworthy as it is the first study (to my knowledge) to characterize bone marrow B cell progenitors using a combination of scRNAseq and scProteomics. This data set is an important reference point for the community and reveal novel aspects of the B cell developmental pathway. the only other paper I know of to use scRNA-seq on (some) comparable populations is Miyai, et al. doi: 10.1101/gad.309575.117 - which should be cited. The submitted paper is an important step towards further refinement of the B cell development pathway. Previous work uses FACS sorting to define populations and then measures/counts transcripts within these populations. scRNAseq paired with scProteomics is an important and sophisticated tool to interrogate the heterogeneity of populations as well as to understand, on a cell-to-cell level, dynamics when specific populations are compared (using both "biased" and unbiased approaches). The conclusions that are made in the paper are well supported by the data.

the authors should address the following before publication:

1. the flow cytometry and FACS methods should be clarified. the basis of gating decisions should be documented (in supplemental data). For analysis, what method was used to exclude doublets from the presented data ?
2. For the single cell separation used to isolate cells for the scRNAseq and scProteomics- there is no documentation as to the method used. was it FACS ? was it DropSeq? if FACS, was index sorting used? what steps were taken to verify that only single cells were isolated?
3. for the populations that were defined - how were they defined ? there is nothing clear about the methods used
4. how can the reader have confidence that each population is rigorously defined ? there seems like an opportunity lost here to test whether previous clusters of proteins used for FACS actually are a homogenous or heterogenous population. this is critical as cells that are inappropriately included dilute the accuracy of quantitative analysis of transcripts and proteins. perhaps <https://www.ncbi.nlm.nih.gov/pmc/articles/PMC5669064/> could be a guiding precedent here.
5. what is the rationale for examining B220+CD43+CD19+CD98+ cells ?
6. B220+CD19+ CD43+ is not a sufficient definition of pro-B cells. first, a number of B220+CD43+ cells are not B-lineage cells, so other markers are usually included, and its essential to exclude doublets in this analysis. also, mature B cells (particularly B1 cells) can be B220+CD43+, so how were these excluded? can this be done using the gene expression and proteomics data ?
7. the traditional Hardy Fraction A is not a rigorous definition of pre-pro-B cells - can the gene expression and proteomics data be used to examine which cells might be excluded based on characterization of B-cell biased/primed but not committed progenitors by others ?
8. the statistical analyses and other approaches to data analyses seem appropriate and rigorous

I support the concept a transparent and open review process and am comfortable signing my name - Rachel Gerstein, PhD,

Reviewer #2 (Remarks to the Author):

The authors sought to use single-cell transcriptomics and proteomics to refine the pre-B cell expansion stages in mouse B cell differentiation and use this knowledge to point to key developmental nodes linked to B cell malignant transformation. Whilst the study is a neat descriptive study of B cell differentiation, the conclusions drawn about B-ALL are so far analytically

premature and need further work.

Major comments:

1. Line 80: the cells separate significantly by cell cycle, which is not expected for a population of cells, and suggests that correction for cell cycle has not been applied to the data in an adequate way to group cells together on the basis of non-cell-cycle associated genes. Even though S/G2M phases are associated with proliferation, we would expect that cells in G1 phase would also be present in any given group of proliferating cells (through sampling by chance). Please can the authors address this major issue, as all downstream clustering relies on this.
2. Whilst there are many similarities between the mouse and human B cell populations, there are also many significant differences, including within some of the major cell subsets. For example, the B1 and B2 cell definitions in mice are not comparable (and still controversial) in humans. Therefore, caution needs to be made when making direct inferences between species. This is particularly important when making assumptions about B cell differentiation in humans derived from mouse models. The authors should discuss this.
3. Line 154: "(LMHH and MAF, submitted)" When the authors refer to data that is submitted but not peer reviewed or published, this cannot be used as a citation. I therefore suggest that the data that would provide confidence in the results of Figure 3F are made available in this manuscript (and can be referenced in the other paper if needed). This may be an editorial decision ultimately, but it runs the risk of citing experiments that may never be published.
4. Figure 7 and the B-ALL analyses: whilst the analyses are described in the methods, it would be useful to have a more thorough description in the Figure legends showing what the B-ALL subgroups represent, and perhaps a table of the subgroups that the authors have identified.
5. B-ALL paragraph: Whilst this analysis shows the heterogeneity of B-ALL subtypes (albeit very little information about the numbers of patients used in this analysis, the exclusion. Inclusion criteria, and covariates used), the results do not directly show "B cell gene expression networks that are modulated during B cell development ... are exploited by human B-ALLs" (line 266). Rather, these gene expression patterns may be a bystander/carrier changes as a result of leukaemic transformation. The authors need to show that that these gene pathways are associated with transformation and/or improved survival/proliferation to be able to make the assertion that these gene networks are "exploited". Secondly, it would be more convincing if the full gene networks were tested as an association with B-ALL rather than just individual genes. I think the authors need to extend this section analytically before drawing such conclusions (albeit an interesting start).

- Appropriate use of statistics and treatment of uncertainties

Yes

- References: appropriate credit to previous work?

These seem appropriate.

- Minor comments

1. The authors do not refer to the panels in Figure 4 in the correct order. Labelling needs to be rearranged.
2. As a stylistic comment, the discussion is very much a repeat of the results. It would serve the paper better to condense the key findings and implications in the discussion only.

Reviewer #3 (Remarks to the Author):

This manuscript by Farrar and coworkers describes a single-cell RNASeq analysis of bone marrow development of mouse B cells. The authors isolate bone marrow cells from two mice, label them with hashtag, CITE-seq and cell sorting antibodies, isolate identical numbers of B220+CD43+ and

B220+CD43⁻ cells and subject them to single cell RNASeq using a 3' 10x system. They define novel differentiation stages, and the CITE-Seq antibodies allows them to compare these stages to those previously defined using flow cytometry. They find that the transcription factors EBF1 is downregulated and Myc upregulated during the novel PreBCRd stage, and that this is associated with changes in expression of genes under control by these. They define further differences in gene expression between stages during preBCRd and preBCRi stages. Using pseudotime analysis, they define gene modules that specifically changes during differentiation. Finally, they compare the expression of genes that they have defined as typical for certain developmental stages to the expression of genes in a panel of B-ALL.

Overall, it is a nicely presented study, and it is the first single-cell RNASeq paper study, as far as I am aware, that focuses entirely on early B cell development in mouse bone marrow. While many of the observations in the manuscript confirms previous observations, the authors also define novel stages of B cell development and gene signatures associated with these. As these are critical steps in the development and selection of B cells expressing diverse antibody receptors, novel stages during B cell development is of general interest.

One drawback of the study is that it in fact describe a single 10X single-cell RNASeq experiment performed in a single 10X well. RNASeq experiments are usually rather robust, and the two mice included in the same well give very similar results. However, it still raises some concerns with regard to how repeatable some of the findings are and also if the data will be seen as a conclusive resource or not.

Another drawback is that the authors do not present any way of isolating or detect cells within the predicted stages using other experimental systems. Ideally, the presence of the preBCRd stage should have been demonstrated using flow cytometry, which would have made it easier for other researchers to build further on the data presented here. This could, for example, be based on expression of CD79 or IL7r on the cell surface, lack of EBF1 using intracellular staining and possibly be coupled to cell cycle stage of the cells. Similarly, the expression CXCR4, CD74 or CD44 could have been used to address preBCRd stages. If good markers are defined, this would also have allowed sorting of cells to confirm observed gene expression differences between stages.

Related to this, while the authors do perform some mouse experiments aimed at defining functions based on their findings, these are not really supporting any importance in B cell development. Thus, while three potential EBF1 target genes are upregulated in light chain-negative early B cells in EBF1/Pax5 haplodeficient mice (although less than two-fold), this only confirms them as likely EBF1 target genes but not as regulated during the PreBCRd stage. And mice deficient for another gene defined as expressed during this developmental stage, YBX3, do not show any changes in B cell development due to its absence, something that authors argue, but do not test, is due to redundancy with YBX1.

Finally, with several of the described stages only consisting of S and G2/M cells, it is hard to understand the correlations in time between the different stages. It would rather seem that cells are going through different types of proliferation in PreBCRd, PreBCRi I and preBCRi II, but that they must then go into a G1 phase, probably reflected by the ProB VDJ or Kappa Pre B, in between these cycles.

Specific comments

1) The authors label the cells with antibodies before sorting, and then sort cells based on expression of B220 and lack of lineage markers. B220 is known to be low on cells within the B1 lineage, possibly even during their early development in bone marrow, and at least one of the lineage markers, CD11c, is known to be expressed on certain peripheral B cells (ABC (Rubtsov, Blood 118: 1305) and GC (Baumjohann, Immunity 38: 596) B cells). I am not aware of any data supporting B cell lineage expression of CD11c in bone marrow, but the expression of B220 is indeed very significantly based on the CITE-Seq antibodies (Figure 1D). One wonders whether some B lineage cells may be lost during sorting. At the least, the authors should demonstrate that the markers used are indeed labelling all known B cells in the bone marrow, comparing their expression to other B cell markers such as CD19 (will miss pre-pro B cells, Fig 1d) and CD79a

(which may also will probably miss pre-pro B cells, Fig 2a) appears promising in that are expressed during most of B cell development in bone marrow, and ensure that no cells are lost due to using B220 or due to excluding CD11c expressing cells.

2) In the material and methods section the authors state that 966 GEMs contained doublets and 7454 GEMs contained single cells. As it is discussed in relation to the hashtagging I presume that it is based on the number of cells having both hashtags rather than the ones excluded due to high expression levels in Seurat. Regardless, this is higher than would be expected by chance alone for this number of analyzed cells, and could be due to that i) there is some leakage of hashtags between the two mice, or that ii) there was cell clumping. The first case will not really influence the data quality, while the second indicates that a similar frequency of doublets will occur within samples (i.e. that more than 10% of all events are doublets). To exclude this possibility the authors should provide a comparison map of the number of hashtags detected on the "doublets". If there is leakage between cells, it would be expected that one hashtag will dominate on most cells, if there is clumping similar numbers of the two hashtags will be detected for each doublet.

3) I do not really understand how the authors imagine that the pre-BCRd and pre-BCRi populations are connected to each other and other populations. A full cell cycle should include all three stages (G1, S and G2/M), and yet these populations are made up of cells in S and G2/M only. Do cells only go through one division while belonging to these populations and then enter into another of the defined populations before taking further loops of cell cycles in the same or other populations? Or is this a cell cycle program unique to early B cell development with extremely short G1 stages? This needs to be clarified, as it is really hard to understand the order of events that the authors imagine during B cell development from the current manuscript. Based on the Monocle UMAP with cell cycles regressed out, it would seem that at least PreBCRi I and II could cycle with Kappa PreB as their G1 stage, which also would explain why the authors see kappa and RAG expression in the cycling cells (Figure 5a).

4) Some efforts need to be done to identify the subsets using normal flow cytometry. Especially the PreBCRd stage should be rather unique based on several of the markers described, including CD79, IL7r and possibly other cell surface markers. Additionally, direct phospho-flow to strengthen the point about PreBCR signals should be attempted.

5) I do not understand if the cell populations imaged in the monocle UMAP are the same clusters defined using the Seurat workflow or are in fact new clusters that are named in accordance with the Seurat clusters. It seems that the later is the case, given that two earlier clusters (PreBCRi II S and G2/M) are joined to one. If this is the case I would like a representation showing the Monocle UMAP with the cells colored in accordance with which Seurat cluster they belong, and a description of how many cells that are classified within the same clusters or within different clusters when the Seurat and Monocle workflows are used.

6) The B-ALL analysis is interesting and contributes to making the study more interesting. However, I do not really understand how the gene sets are defined. The authors say that they were identified in clusters using Seurat and Monocle (the intersection between the two) but not how these genes were first identified (genes unique to one subset compared to all cells or compared to its neighbors, how many genes and what was required to be included (relative upregulation, P value, are all genes reaching this threshold included) and what proportion that was shared between Seurat and Monocle). Also – in the cycling populations – genes associated with cell cycle should be marked. Since there likely will be cycling ALL cells, there could be expression of cell cycle genes that are not strictly due to cell stage but only cell cycle.

Minor points

7) The use of similar colors for neighboring populations is visibly pleasing but makes it hard to distinguish populations from each other.

8) Some of the text in the figures is really small (in particular Figure 3E) and close to impossible to see.

9) Mki67 in Figure 1C should be in bold if it should be similar to other headlines.

10) I think there a headline for Figure 7A in similar with the ones on Supplemental figure 6 that state that this is genes considered unique to the PreBCRd population should be included. Also – why are not gene sets defined from all populations but only selected ones in Sup Fig 6. The analysis is now done for 9 our of 13 populations – it seems strange that the last four are left out.

Response to Review

The reviewer's critique of our manuscript and our response to the reviewer's concerns (in red) are outlined below. Changes to the actual manuscript are marked in yellow highlighting.

Reviewer #1 (Remarks to the Author):

This study used single-cell transcriptomics and proteomics to characterize B cell development. The analysis revealed unique transcriptional signatures that characterize the pre-B cell expansion stages into novel pre-BCR-dependent and pre-BCR-independent phases. These changes correlate with unexpected dynamic changes in expression of the transcription factor EBF1 and the RNA binding protein YBX3, that are defining features of the pre-BCR-dependent stage. Detailed analyses also characterize the expression kinetics ..across B cell development of transcription factors, cytokines, chemokines, and their associated receptors. An important set of findings in this Ms point to key developmental nodes linked to B cell transformation (ALL).

the paper is noteworthy as it is the first study (to my knowledge) to characterize bone marrow B cell progenitors using a combination of scRNAseq and scProteomics. This data set is an important reference point for the community and reveal novel aspects of the B cell developmental pathway. the only other paper I know of to use scRNA-seq on (some) comparable populations is Miyai, et al. doi: 10.1101/gad.309575.117 - which should be cited. The submitted paper is an important step towards further refinement of the B cell development pathway. Previous work uses FACS sorting to define populations and then measures/counts transcripts within these populations. scRNAseq paired with scProteomics is an important and sophisticated tool to interrogate the heterogeneity of populations as well as to understand, on a cell-to-cell level, dynamics when specific populations are compared (using both "biased" and unbiased approaches). The conclusions that are made in the paper are well supported by the data.

the authors should address the following before publication:

1. the flow cytometry and FACS methods should be clarified. the basis of gating decisions should be documented (in supplemental data). For analysis, what method was used to exclude doublets from the presented data ?

We now include a figure showing how cells were gated using flow cytometry. It is important to point out that flow cytometry was largely used to focus on B cells (although possibly lacking B1 B cell progenitors which may lack B220 expression) and enrich for the early stages of B cell development so that we would have enough pre-proB and pro-B cells for analysis. Doublets during flow sorting were excluded using forward and side scatter gates. More importantly, we were able to remove doublets that were also captured in the same GEMs during the scRNA-Seq analysis. This study was part of a larger project in which we combined cells from WT and two distinct KO mice. Each genotype was represented in duplicate for a total of 6 mice. Cells from each mouse were labeled with a unique hashtag antibody. The vast majority of doublets (>83% will

express two distinct hashtags) thereby allowing us to remove most doublets. A figure showing this is provided here for the reviewers.

Left panel, shows the relative distribution of cells among clusters containing a single hashtag (blue) versus those that has two or more hashtags (red). Right panel; Identical plot just showing the breakdown of which clusters expressed which set of hashtags. For the analysis in this paper, we focused only on single cells associated with WT1 and WT2 mice. The clustering shown above is entirely based on hashtag distribution, not transcriptome, so one should not make inferences about how similar transcriptomes are based on differences (ie, even though WT1 and WT2 are far apart this is based on their hashtag names not their transcriptomes).

2. For the single cell separation used to isolate cells for the scRNAseq and scProteomics- there is no documentation as to the method used. was it FACS ? was it DropSeq? if FACS, was index sorting used? what steps were taken to verify that only single cells were isolated?

The method used was 10xgenomics in which individual cells are partitioned into single droplets. Using both hashtag antibodies and software algorithms we can eliminate almost all droplets that contained more than one cell.

3. for the populations that were defined - how were they defined ? there is nothing clear about the methods used

The process used to pick a specific resolution is somewhat iterative. We essentially increased the resolution at which we defined subsets until further subdivisions failed to yield clearly distinct cells subsets. An example of this is shown here.

Increasing numbers of clusters as resolution increases. At the lowest resolution (0.05), there are four main clusters corresponding to pre-pro-B, pro-B, pre-B and immature/mature B cells. As we increase the resolution, we see pro-B cells split into VDJ rearranging versus cycling and immature versus mature B cells split (0.1), pre-BCRd versus pre-BCRi split (0.15), pre-BCRi split (0.2), kappa versus lambda split (0.3), and pre-BCRi II split (0.4).

The robustness of the clusters was further validated using the pairwise cluster differential expression list. Any pairwise cluster comparison that had fewer than 5 differentially expressed genes were merged into a newly labeled joint cluster, and the process of pairwise DE testing would begin again using the revised cluster classifications. At the final cluster resolution of 0.4, there were zero cluster merging steps, suggesting robust clustering. We have added this information to the methods section.

4. how can the reader have confidence that each population is rigorously defined ? there seems like an opportunity lost here to test whether previous clusters of proteins used for FACS actually are a homogenous or heterogenous population. this is critical as cells that are inappropriately included dilute the accuracy of quantitative analysis of transcripts and proteins.

perhaps <https://www.ncbi.nlm.nih.gov/pmc/articles/PMC5669064/> could be a guiding precedent here.

Clustering of populations using Seurat is by nature an empirical process. By increasing the resolution we can continuously subdivide populations. Determining whether a new subdivision represents a distinct B cell subset or not is not always trivial. In the case of pre-B cells, we were able to clearly identify new subsets that we now term pre-BCR-dependent (pre-BCRd) and Pre-BCR independent (Pre-BCRi). Furthermore, our data allows us to clearly distinguish immature B cell development into distinct kappa and lambda rearranging subsets. As the reviewer points out, our data might define other subsets as well. Identifying all potential subsets is beyond the scope of this single paper, although our data should be a useful resource that will allow other investigators to explore these questions in greater detail. In addition, our studies will provide a

resource that will allow us and others to explore the underlying mechanisms underlying developmental transitions. We provide a start here by identifying new modules that most clearly define distinct B cell subsets and show how this can be applied to better characterize B cell leukemias.

5. what is the rationale for examining B220+CD43+CD19+CD98+ cells ?

This description in part relates to the previous question. As part of our studies of the pre-BCR dependent subset we identified transcripts for the neutral amino acid transporters (CD98) as a defining feature of the pre-BCRd stage of B cell development. These studies show that we can also identify expression of CD98 by flow cytometry. Per suggestions of the reviewers we now include other cell surface markers (CD74, IL7R, CXCR4 and MYC (based on *Myc-GFP* reporter mice) to better define the pre-BCRd and pre-BCRi subsets.

6. B220+CD19+ CD43+ is not a sufficient definition of pro-B cells. first, a number of B220+CD43+ cells are not B-lineage cells, so other markers are usually included, and its essential to exclude doublets in this analysis. also, mature B cells (particularly B1 cells) can be B220+CD43+, so how were these excluded? can this be done using the gene expression and proteomics data ?

We sorted on B220+CD19+CD43+ cells not to restrict these cells just to the progenitor B cell subsets, but mainly to enrich for these subsets as otherwise they would comprise only a tiny fraction of our single cell analysis. Subsequent analysis of RNA transcripts and the use of CITE-seq antibodies allow us to distinguish between progenitor B cells and other developing B cell subsets as well as non-B cell lineages. For example, the CITE-Seq antibody to the cell surface marker AA4.1 should distinguish mature from immature B cell subsets. We did indeed find B220+CD43+ mature B cells but noted that they expressed other markers, such as ApoE, that distinguished them from progenitor B cells. Likewise, B lineage pre-pro-B cells could be distinguished from other lineages that fall within Hardy fraction A via AA4.1 expression. Doublet exclusion is described above.

7. the traditional Hardy Fraction A is not a rigorous definition of pre-pro-B cells - can the gene expression and proteomics data be used to examine which cells might be excluded based on characterization of B-cell biased/primed but not committed progenitors by others ?

We used the AA4.1 CITE-Seq antibody to further refine the pre-proB cell subset. These data demonstrate that the AA4.1+ cells are enriched in those expressing *Ebf1*, *Vpreb1* and *Vpreb3*, and *CD24a* further confirming that these are the true pre-proB cell progenitors. These data are now shown in figure 2D.

8. the statistical analyses and other approaches to data analyses seem appropriate and rigorous

I support the concept a transparent and open review process and am comfortable signing my name - Rachel Gerstein, PhD,

Reviewer #2 (Remarks to the Author):

The authors sought to use single-cell transcriptomics and proteomics to refine the pre-B cell expansion stages in mouse B cell differentiation and use this knowledge to point to key developmental nodes linked to B cell malignant transformation. Whilst the study is a neat descriptive study of B cell differentiation, the conclusions drawn about B-ALL are so far analytically premature and need further work.

Major comments:

1. Line 80: the cells separate significantly by cell cycle, which is not expected for a population of cells, and suggests that correction for cell cycle has not been applied to the data in an adequate way to group cells together on the basis of non-cell-cycle associated genes. Even though S/G2M phases are associated with proliferation, we would expect that cells in G1 phase would also be present in any given group of proliferating cells (through sampling by chance). Please can the authors address this major issue, as all downstream clustering relies on this.

Since B cell cycle is such a predominant and defining feature of B cell differentiation, we did not want to exclude cell cycle genes from the initial Seurat based analyses shown in figure 1. We did regress out cell cycle genes for the subsequent monocle analysis shown in figure 7. Thus, analyses with and without cell cycle regression were carried out.

The second question raised here is an excellent question and one we have pondered as well. A recent study by Ranjan Sen and colleagues (Bioarxiv doi:<https://doi.org/10.1101/796904>) provides a potential explanation. These authors found that rapidly proliferating splenic B cells go through a distinct cell cycle process in which genes associated with G2M are not fully extinguished during G1, and thus cells go through an abbreviated cell cycle with an extremely short non-canonical G1 stage that bears features of G2/M. This likely holds true for rapidly proliferating pre-B cells as well. Since the cell cycle algorithm we used tries to best match transcriptome to a specific stage it would likely classify these hybridG2M/G1 cells as G2M cells. We now describe this possibility in the discussion.

2. Whilst there are many similarities between the mouse and human B cell populations, there are also many significant differences, including within some of the major cell subsets. For example, the B1 and B2 cell definitions in mice are not comparable (and still controversial) in humans. Therefore, caution needs to be made when making direct inferences between species. This is particularly important when making assumptions

about B cell differentiation in humans derived from mouse models. The authors should discuss this.

We acknowledge that while many aspects of murine and human B cell differentiation are highly analogous, they are not identical in all aspects. We have added a statement to this effect in the discussion.

3. Line 154: "(LMHH and MAF, submitted)" When the authors refer to data that is submitted but not peer reviewed or published, this cannot be used as a citation. I therefore suggest that the data that would provide confidence in the results of Figure 3F are made available in this manuscript (and can be referenced in the other paper if needed). This may be an editorial decision ultimately, but it runs the risk of citing experiments that may never be published.

This manuscript is currently under revision at Oncogene. This manuscript was resubmitted two weeks ago and will hopefully be in press soon. In the meantime, the manuscript was submitted to Biorxiv (doi: <https://doi.org/10.1101/2020.11.25.398966>) and is available there for readers to examine. We have now included this reference in the manuscript.

4. Figure 7 and the B-ALL analyses: whilst the analyses are described in the methods, it would be useful to have a more thorough description in the Figure legends showing what the B-ALL subgroups represent, and perhaps a table of the subgroups that the authors have identified.

This is a good point. To try to simplify the visualization of this data, we have now reduced the gene sets describing each gene module to z-scores and now present a figure showing how the distinct gene modules correlate with distinct types of human B cell leukemia. This data is now presented in figure 7a.

5. B-ALL paragraph: Whilst this analysis shows the heterogeneity of B-ALL subtypes (albeit very little information about the numbers of patients used in this analysis, the exclusion. Inclusion criteria, and covariates used), the results do not directly show "B cell gene expression networks that are modulated during B cell development ... are exploited by human B-ALLs" (line 266). Rather, these gene expression patterns may be a bystander/carrier changes as a result of leukaemic transformation. The authors need to show that that these gene pathways are associated with transformation and/or improved survival/proliferation to be able to make the assertion that these gene networks are "exploited". Secondly, it would be more convincing if the full gene networks were tested as an association with B-ALL rather than just individual genes. I think the authors need to extend this section analytically before drawing such conclusions (albeit an interesting start).

This question relates in part to the previous one. The identification of gene modules that correlate to individual leukemias should simplify this data. We acknowledge that at this

point our data show correlation of these genes modules, thus we have changed the wording from “exploited” by human B-ALL to “associate” with human B-ALL.

- Appropriate use of statistics and treatment of uncertainties

Yes

- References: appropriate credit to previous work?

These seem appropriate.

- Minor comments

1. The authors do not refer to the panels in Figure 4 in the correct order. Labelling needs to be rearranged.

We have corrected the order in which these subpanels are called out.

2. As a stylistic comment, the discussion is very much a repeat of the results. It would serve the paper better to condense the key findings and implications in the discussion only.

Based on the reviewers comments we have removed some of the repetitive items and replaced them with issues related to specific concerns of the reviewers (for example, discussion of the cell cycle, etc.).

Reviewer #3 (Remarks to the Author):

This manuscript by Farrar and coworkers describes a single-cell RNASeq analysis of bone marrow development of mouse B cells. The authors isolate bone marrow cells from two mice, label them with hashtag, CITE-seq and cell sorting antibodies, isolate identical numbers of B220+CD43+ and B220+CD43- cells and subject them to single cell RNASeq using a 3' 10x system. They define novel differentiation stages, and the CITE-Seq antibodies allows them to compare these stages to those previously defined using flow cytometry. They find that the transcription factors EBF1 is downregulated and Myc upregulated during the novel PreBCRd stage, and that this is associated with changes in expression of genes under control by these. They define further differences in gene expression between stages during preBCRd and preBCRi stages. Using pseudotime analysis, they define gene modules that specifically changes during differentiation. Finally, they compare the expression of genes that they have defined as typical for certain developmental stages to the expression of genes in a panel of B-ALL.

Overall, it is a nicely presented study, and it is the first single-cell RNASeq paper study, as far as I am aware, that focuses entirely on early B cell development in mouse bone marrow. While many of the observations in the manuscript confirms previous observations, the authors also define novel stages of B cell development and gene signatures associated with these. As these are critical steps in the development and selection of B cells expressing diverse antibody receptors, novel stages during B cell development is of general interest.

One drawback of the study is that it in fact describe a single 10X single-cell RNASeq experiment performed in a single 10X well. RNASeq experiments are usually rather robust, and the two mice included in the same well give very similar results. However, it still raises some concerns with regard to how repeatable some of the findings are and also if the data will be seen as a conclusive resource or not.

For these studies we have focused on biological replicates which typically encompass more variability than technical replicates. As shown in supplemental figure1, the independent biological replicates were strongly concordant. In our hands carrying out multiple single cell studies of T cells, we have found that we do not find significant differences between subsets identified in independent single cell datasets. The strong concordance with previously published flow cytometry data further suggest that technical variation should not be a major source of error.

Another drawback is that the authors do not present any way of isolating or detect cells within the predicted stages using other experimental systems. Ideally, the presence of the preBCRd stage should have been demonstrated using flow cytometry, which would have made it easier for other researchers to build further on the data presented here. This could, for example, be based on expression of CD79 or IL7r on the cell surface, lack of EBF1 using intracellular staining and possibly be coupled to cell cycle stage of the cells. Similarly, the expression CXCR4, CD74 or CD44 could have been used to address preBCRd stages. If good markers are defined, this would also have allowed sorting of cells to confirm observed gene expression differences between stages.

To address this point, we have carried out flow cytometry studies using the markers requested. We found that CD74 had a very interesting expression pattern, with lower transcripts in pre-BCRd versus Pre-BCRi stages. Furthermore, transcript abundance increased dramatically in immature and mature B cells but surface expression of CD74 decreased. This difference is accounted for by the observation that intracellular levels of CD74 accumulate at the immature B cell stage. Our findings suggest that CD74 is expressed on the surface during the pre-B cell stage, when it is expected to act as co-receptor for the chemokine MIF, but that it relocates to the cytoplasm later in B cell development, where it plays its more classical role as invariant chain involved in MHCII antigen presentation. To further explore this, we made use of *Myc-GFP* reporter mice to better identify the pre-B cell subsets. MYC-GFP-expressing pre-B cells showed heterogenous expression of EBF1 with some cells expressing significantly lower levels of EBF1 protein, thereby confirming the dynamic changes in *Ebf1*/EBF1 expression during pre-B cell differentiation. Using this tool, we further demonstrated staining for

CXCR4, and IL7R as requested. These data show that expression of those two factors and CD74 correlated with EBF1 expression as predicted by the scRNA-Seq data.

Related to this, while the authors do perform some mouse experiments aimed at defining functions based on their findings, these are not really supporting any importance in B cell development. Thus, while three potential EBF1 target genes are upregulated in light chain-negative early B cells in EBF1/Pax5 haplodeficient mice (although less than two-fold), this only confirms them as likely EBF1 target genes but not as regulated during the PreBCRd stage. And mice deficient for another gene defined as expressed during this developmental stage, YBX3, do not show any changes in B cell development due to its absence, something that authors argue, but do not test, is due to redundancy with YBX1.

We have tried to test potential redundancy of YBX1 and YBX3. We were able to use CRISPR to knock-out *Ybx1* or *Ybx3* in a human Nalm6 pre-B leukemia cell line but we were not able to generate validated *Ybx1/3* DKO. Thus, this technical failure, despite repeated attempts, precludes us from definitively addressing this question.

Finally, with several of the described stages only consisting of S and G2/M cells, it is hard to understand the correlations in time between the different stages. It would rather seem that cells are going through different types of proliferation in PreBCRd, PreBCRi I and preBCRi II, but that they must then go into a G1 phase, probably reflected by the ProB VDJ or Kappa Pre B, in between these cycles.

It is possible that there is a switch between cycling proB VDJ rearrangement. We suspect this is not the case as the major proliferative burst appears to be driven by a functionally rearranged heavy chain. As described for reviewer 2 above, we believe our findings may reflect recent work by Sen and colleagues showing that rapidly proliferating B cells lack a phenotypically distinct G1 stage (see BioRxiv doi:<https://doi.org/10.1101/796904>).

Specific comments

1) The authors label the cells with antibodies before sorting, and then sort cells based on expression of B220 and lack of lineage markers. B220 is known to be low on cells within the B1 lineage, possibly even during their early development in bone marrow, and at least one of the lineage markers, CD11c, is known to be expressed on certain peripheral B cells (ABC (Rubtsov, Blood 118: 1305) and GC (Baumjohann, Immunity 38: 596) B cells). I am not aware of any data supporting B cell lineage expression of CD11c in bone marrow, but the expression of B220 is indeed very significantly based on the CITE-Seq antibodies (Figure 1D). One wonders whether some B lineage cells may be lost during sorting. At the least, the authors should demonstrate that the markers used are indeed labelling all known B cells in the bone marrow, comparing their expression to other B cell markers such as CD19 (will miss pre-pro B cells, Fig 1d) and CD79a (which may also will probably miss pre-pro B cells, Fig 2a) appears promising in that are expressed during most of B cell development in bone

marrow, and ensure that no cells are lost due to using B220 or due to excluding CD11c expressing cells.

As the reviewer points out, our sort strategy may not capture all B cell subsets, most notably B220-CD19+ B1 B cell progenitors. We now mention this in the discussion.

2) In the material and methods section the authors state that 966 GEMs contained doublets and 7454 GEMS contained single cells. As it is discussed in relation to the hashtagging I presume that it is based on the number of cells having both hashtags rather than the ones excluded due to high expression levels in Seurat. Regardless, this is higher than would be expected by chance alone for this number of analyzed cells, and could be due to that i) there is some leakage of hashtags between the two mice, or that ii) there was cell clumping. The first case will not really influence the data quality, while the second indicates that a similar frequency of doublets will occur within samples (i.e. that more than 10% of all events are doublets). To exclude this possibility the authors should provide a comparison map of the number of hashtags detected on the “doublets”. If there is leakage between cells, it would be expected that one hashtag will dominate on most cells, if there is clumping similar numbers of the two hashtags will be detected for each doublet.

In response to this query and that of reviewer 1, we now show how hashtagging eliminates the vast majority of doublets. In addition, we used Seurat based algorithms described in the methods that further limit the presence of doublets. This is further described in the methods. Thus, while doublets can never be completely excluded using 10x gem-based methods, we can limit them to a very small fraction of the total events identified.

3) I do not really understand how the authors imagine that the pre-BCRd and pre-BCRi populations are connected to each other and other populations. A full cell cycle should include all three stages (G1, S and G2/M), and yet these populations are made up of cells in S and G2/M only. Do cells only go through one division while belonging to these populations and then enter into another of the defined populations before taking further loops of cell cycles in the same or other populations? Or is this a cell cycle program unique to early B cell development with extremely short G1 stages? This needs to be clarified, as it is really hard to understand the order of events that the authors imagine during B cell development from the current manuscript. Based on the Monocle UMAP with cell cycles regressed out, it would seem that at least PreBCRi I and II could cycle with Kappa PreB as their G1 stage, which also would explain why the authors see kappa and RAG expression in the cycling cells (Figure 5a).

This is a very intriguing question that was also brought up by reviewer 2 and one that we also find interesting. We do not believe that cells only go through one division before leaving the pre-BCRd /pre-BCRi-dependent stages. As the reviewer suggests a more likely explanation is a cell cycle program featuring extremely short G1 stages. We do not believe this is unique to early progenitor B cells as a similar observation was

recently proposed for proliferating germinal center B cells. Specifically, a recent study by Ranjan Sen and colleagues (Bioarxiv doi:<https://doi.org/10.1101/796904>) provides a potential explanation for this question. These authors found that rapidly proliferating splenic B cells go through a distinct cell cycle process in which genes associated with G2M are not fully extinguished during G1, and thus cells go through an abbreviated cell cycle with an extremely short non-canonical G1 stage that bears features of G2/M. This likely holds true for rapidly proliferating pre-B cells as well. Since the cell cycle algorithm we used tries to best match transcriptome to a specific stage it would likely classify these hybrid G2M/G1 cells as G2M cells. We now describe this possibility in the discussion.

4) Some efforts need to be done to identify the subsets using normal flow cytometry. Especially the PreBCRd stage should be rather unique based on several of the markers described, including CD79, IL7r and possibly other cell surface markers. Additionally, direct phospho-flow to strengthen the point about PreBCR signals should be attempted.

We have carried out the requested flow cytometry studies and now include them in supplementary figure 4.

5) I do not understand if the cell populations imaged in the monocle UMAP are the same clusters defined using the Seurat workflow or are in fact new clusters that are named in accordance with the Seurat clusters. It seems that the later is the case, given that two earlier clusters (PreBCRi II S and G2/M) are joined to one. If this is the case I would like a representation showing the Monocle UMAP with the cells colored in accordance with which Seurat cluster they belong, and a description of how many cells that are classified within the same clusters or within different clusters when the Seurat and Monocle workflows are used.

The monocle analysis was labeled according to the Seurat defined clusters (panel 7A) and does not show new clusters. However, in the monocle analysis we regressed out cell cycle as cell cycle would otherwise dominate this process and inappropriately juxtapose cycling pro-B cells and cycling pre-B cells.

6) The B-ALL analysis is interesting and contribute to making the study more interesting. However, I do not really understand how the gene sets are defined. The authors say that they were identified in clusters using Seurat and Monocle (the intersection between the two) but not how these genes were first identified (genes unique to one subset compared to all cells or compared to its neighbors, how many genes and what was required to be included (relative upregulation, P value, are all genes reaching this threshold included) and what proportion that was shared between Seurat and Monocle). Also – in the cycling populations – genes associated with cell cycle should be marked. Since there likely will be cycling ALL cells, there could be expression of cells cycle genes that are not strictly due to cell stage but only cell cycle.

We have clarified how the gene modules were identified and how z-scores for new figure 7a were calculated to clarify these points.

The `findAllMarkers()` function from Seurat was used as described in the **Single-cell Bioinformatic Analyses** to identify which genes are differentially expressed in one cluster compared with all other clusters for the Seurat cluster. The `top_markers()` function from Monocle was used to identify which genes are differentially expressed in one cluster compared with all others for the Monocle cluster definitions. We used the Monocle default parameters, so 25 genes were returned per Monocle cluster with the `top_markers()` function. We did not apply any additional filters to the top gene markers for the Monocle analysis. For the Seurat cluster marker genes we only used genes with positive log fold changes to better match the Monocle cluster gene marker results. We did not include downregulated marker genes from the Seurat analysis. For both Monocle and Seurat cluster gene lists we had to convert the mouse gene names to human gene names, so some genes were removed that did not have appropriate human homologs. Additionally, when there wasn't a direct 1:1 mapping between the Monocle and Seurat clusters, cluster gene lists from one method were merged before comparison to the other method's cluster gene list. We then used the intersection of the Monocle lists and the Seurat lists to get our final list of genes for each cluster for our individual B-ALL heatmaps for each group of cluster marker genes.

To determine the association between gene networks and the different human B-ALL subtypes, we calculated each cluster gene set z-score by averaging the z-scores of the cluster marker genes in each individual cluster gene set for each B-ALL subtype. In the heatmap of these data we allowed for both column (B-ALL subtype) and row (average cluster z-score) hierarchical clustering. The overlap for each cluster between Monocle and Seurat are outlined below.

Overlap for cluster: Kappa Pre B
Intersect Length: 16
Monocle list length: 26
Seurat list membership: 0.37
Monocle list membership: 0.62

Overlap for cluster: Immature B
Intersect Length: 26
Monocle list length: 43
Seurat list membership: 0.53
Monocle list membership: 0.60

Overlap for cluster: PreBCRi I
Intersect Length: 14
Monocle list length: 14
Seurat list membership: 0.046
Monocle list membership: 1

Overlap for cluster: Mature B
Intersect Length: 19

Monocle list length: 19
Seurat list membership: 0.21
Monocle list membership: 1

Overlap for cluster: Pro B VDJ
Intersect Length: 25
Monocle list length: 25
Seurat list membership: 0.24
Monocle list membership: 1

Overlap for cluster: Lambda Pre B
Intersect Length: 17
Monocle list length: 21
Seurat list membership: 0.53
Monocle list membership: 0.81

Overlap for cluster: Cycling Pro B
Intersect Length: 24
Monocle list length: 24

Seurat list membership: 0.07
Monocle list membership: 1

Overlap for cluster: PreBCRi II S phase
Intersect Length: 22
Monocle list length: 23
Seurat list membership: 0.16
Monocle list membership: 0.96

Overlap for cluster: PreBCRd
Intersect Length: 25
Monocle list length: 25
Seurat list membership: 0.039
Monocle list membership: 1

Overlap for cluster: Prepro B
Intersect Length: 23
Monocle list length: 23
Seurat list membership: 0.077
Monocle list membership: 1

Overlap for cluster: PreBCRi II G2/M
phase

Intersect Length: 12
Monocle list length: 23
Seurat list membership: 0.044
Monocle list membership: 0.52

Overlap for cluster: Cycling Immature B
Intersect Length: 20
Monocle list length: 43
Seurat list membership: 0.087
Monocle list membership: 0.47

Overlap for cluster: Plasma Cells
Intersect Length: 19
Monocle list length: 19
Seurat list membership: 0.12
Monocle list membership: 1

Overlap for cluster: High mitochondrial B
Intersect Length: 18
Monocle list length: 24
Seurat list membership: 0.82
Monocle list membership: 0.75

Minor points

7) The use of similar colors for neighboring populations is visibly pleasing but makes it hard to distinguish populations from each other.

We have recolored the different stages in monocle to better visualize differences between stages.

8) Some of text in the figures is really small (in particular Figure 3E) and close to impossible to see.

We have enlarged the text for key figures to improve clarity.

9) Mki67 in Figure 1C should be in bold if it should be similar to other headlines.

We have changed this to better match other headlines.

10) I think there a headline for Figure 7A in similar with the ones on Supplemental figure 6 that state that this is genes considered unique to the PreBCRd population should be included. Also – why are not gene sets defined from all populations but only selected

ones in Sup Fig 6. The analysis is now done for 9 out of 13 populations – it seems strange that the last four are left out.

We now include all 13 subsets (we are not sure how relevant the high mitochondrial and plasma cell subsets are). Furthermore, we have reduced the genes in each module to z-scores that allow us to present a simplified figure showing how all the different modules correlate with distinct leukemia subsets.

Finally, a log of all changes made to figures is included below for the reviewer's benefit (this should be redundant with what is written above so feel free to ignore it if it is not helpful). Changes to the text of the manuscript are outlined by yellow highlighting.

Change log (Figures)

1. Figure 1. Panel A: Added PacBlue to B220 label and PE to CD43 label
2. Figure 1. Panel C: Bolded Mki67
3. Added Figure 2D. Gene expression from 50-50 median split of CD93hi vs CD93low differential from
4. Increased size of Figure 3E (Ebf1 CHIP-Seq MACS peaks at Nrgn, Slc7a5, and Slc3a2).
5. Figure 4. Moved Panel Figure 4E to Figure 4B due to it being referenced earlier. Subsequently moved Figure 4B, 4C, and 4D into Figure 4C, 4D, and 4E, respectively.
6. Made new Figure 5. Flow cytometric based identification of preBCRd and preBCRi using key markers identified from single-cell RNA-seq
7. Changed old Figures 5,6, and 7 to Figure 6,7, and 8, respectively.
8. Figure 8. Added in cluster vs B-ALL subtype heatmap. Updated heatmap for PreBCRd with bigger labels.
9. Supplementary Figure 3. Added in Fluorophores for each marker (B220-BUV395, CD98-PECy7, Ki67-BV421, JAK1-AF488, EBF1-PE, BP1-BV896, CD24-PerCP-Cy5., B220-BUV395, IgM-APC
10. Supplementary Figure 6. Added all 13 cluster B-ALL heatmaps (excluding preBCRd as that is in main figure)
11. Changed monocle color figures (Figure 7A) and Transcription Factor genes (Figure 7E) and Cytokine genes (Figure 7F)
12. Created Supplementary Figure 1 with flow cytometry gating scheme from the single-cell RNA-seq sorting
13. Changed Figure 3E removed left legends and increased all font size for better readability.
14. Made new summary figure and added into Figure 7D
15. Added Supplementary Figure 5. CD74 spatiotemporal regulation and MycGFP mice analysis.

REVIEWER COMMENTS

Reviewer #1 (Remarks to the Author):

The authors have added new supplementary figures and revised the manuscript - first, these revisions have improved the paper and second, they are responsive to the reviewers. Further revisions are not needed.

This paper reports very interesting results and is a nice step forward for this field

Reviewer #2 (Remarks to the Author):

The authors have made significant efforts to improve their manuscript and have addressed most of the comments well. However, some of the comments have not been tackled properly and need to be improved before publication:

- The authors make no comments in the manuscript about how doublets were removed and this should be included in the methods. In the response to reviewer 1, we commend the authors for using the hashtag-seq information to identify doublets, however, this should be clearly described in the methods, such as how they defined positivity for each sample. The other thing to note is that this will only identify cross-sample doublets, which the authors claim to be only 83% of doublets. Therefore, I recommend another approach to identify these (MLtiplot, DoubletFinder or equivalent may be helpful). This must be addressed.
- Following up from reviewer 1's comments, it would be useful to have a figure of the CITE-seq and GEX signatures of each population so that the readers can be convinced of the groupings (such as a dotplot).
- Figure 7 is still not very informative for a reader. Can the authors indicate which modules are statistically associated with the different leukaemia subtypes? Otherwise, a reader cannot conclude that this is true signal or driven by noise.
- Line 292: "Finally, we examined whether YBX3 expression correlated with outcome in B-ALL": have the authors corrected for multiple testing here? Presumably the authors tested all of the "pre-BCR module genes, including YBX3 and NRG1, [that] are significantly upregulated in BCL2/MYC, IKZF1 N159Y, and MEF2D B-ALL subtypes", and therefore just showing the results for YBX3 may be misleading. Were any of the other genes significantly associated? This needs to be addressed before any conclusions can be made in the manuscript.
- In the rebuttal to reviewers the authors provide the "The overlap for each cluster between Monocle and Seurat" (page 12), however some of the memberships for the Seurat list are very low (<0.05), suggesting poor agreement and low consistency between methods. This does not seem appropriate, and wonder how the analysis would look if the authors correlated Seurat-only derived gene sets and monocle-only derived gene sets?

Reviewer #3 (Remarks to the Author):

My original concerns with this manuscript is listed below with a short description of the response from the authors

Original concern - author response

Only based on two mice from one experiment - the authors have not added additional experiments and argue that the experimental setup is sufficiently robust

No data on protein expression of markers identified as differentially expressed - the authors have added flow cytometry data that support their findings.

Data presented as functional data do not really support their claims - the authors have tried to address if there is redundancy between YBX1 and YBX3 but failed to delete both simultaneously in a B cell line.

The model suggested by the authors would require cells to lack B cells in G1 – the authors links to a pre-print studying proliferation of mature B cells that suggest that B cells may express some G2/M genes even during G1

The potential that distinct B cells subtypes may be lacking due to using B220 and CD11c for cell sorting – this is mentioned in the discussion

The number of doublets seem to be very high compared to what is expected – the authors show that the experiment included more hashtagged samples than the two presented. This explains the inconsistency since doublets counted include all doublets, including those between cells in this analysis and other cells.

Are the clusters in Monocle the same ones as the ones in Seurat? – the authors do not really address this in their responses and are still vague whether cells belonging to a specific Seurat cluster will be kept in that specific cluster in Monocle (i.e. if a specific cell that belonged to a specific Seurat cluster will it keep its color in Monocle), or if the clustering in Monocle of cells is different from the one used in Seurat (i.e. that the cells were reclustered and recolored in Monocle using a similar algorithm as in Seurat but with cell cycles regressed out so that some cells may belong to different clusters in Seurat and Monocle).

The B-ALL analysis is not described in any detail – the authors have improved this significantly in the revised manuscript.

In most cases I think that the authors have addressed my concerns, and that the manuscript have improved in the process.

However, I still have some concerns that I think must be further addressed before publication

Specifically

1) The lack of a G1 phase or not is very important for the conclusion in the manuscript. Many conclusions in the manuscript will change if the cycling cells to the right in the UMAP plot (Cycling ProB, Pro B VDJ, PreBCRd, PreBCRi) are connected to non-cycling or G1 cells ProB VDJ or Kappa Pre B) in the same cell cycle or not. In fact, if they are, this would not really support the model for early B cell differentiation suggested by the authors. I agree with the authors that the preprint to some extent could explain why it is hard to find typical G1 stage B cells among cycling B cells. However, while the preprint suggests some overlap between genes expressed in G2/M and G1 in cycling B cells, possibly due to that rapidly dividing lymphocytes not need additional stimulatory signals in G1 to continue to cycle, this does not mean that cells do not enter G1 phase at all. In fact, as many as 70% of the B cells were classical G1 cells (based on DNA content and DNA synthesis) in the preprint. Rather, the major finding in that report is that there are cells mostly in G1 that express some G2/M phase genes.

I fully appreciate the work that the authors already have put into the analysis of the data in the current manuscript, but it should be relatively easy to address whether individual cells express both G1 and G2/M genes when proliferating or only G2/M genes. At the very least, the authors should look for typical G1 genes among their G2/M class cells at a single cell level and also the combined expression of genes included in their cell cycle analysis (Figure 1C) - do some cells have high values for both G1 and G2M genes within the preBCR clusters? In addition, by isolating the individual cell clusters and then sub-cluster them, possibly while regressing out cell cycle genes, may allow for a higher resolution of the analysis.

I do not find that it is justified to present the current data based on the assumption that bone marrow cells would not have any distinguishable G1 phase - which in fact would be a very major finding. We have found that cell cycle gene expression differences are often dominant when single-cell RNASeq data from cycling cells are clustered, and that this may sometimes hide true differences. I do not think that the authors sufficiently prove that this is not the reason for that some of the different subpopulations presented here are found. This can, and should, be addressed

from existing data.

2) The correlation between clusters in the Seurat and Monocle analysis must be made absolutely clear.

Two of the Seurat clusters (PreBCRi II S phase and PreBCRi II G2/M phase) are presented as one cluster in Monocle (PreBCRi II), and Immature B and Cycling Immature (Seurat) presented as Immature B I and Immature B II (Monocle). Furthermore, statements such as "Using stage-defining markers, we identified 13 distinct B cell developmental stages" and "After adjustment for cell cycle status, UMAP dimensional reduction and clustering were performed in Monocle" suggest that the clusters may not be identical.

If the cells are reclustered in Monocle, after regressing out cell cycle genes, then this must also be made absolutely clear in the text. In addition, and the authors need to present the Monocle dot-plot representation with cells identities based on the Seurat clustering as a Supplementary figure to allow for the reader to compare how well the two methods overlapped with each other in identifying identically named clusters. Without this information it is impossible to determine if there is any correlation between the two methods for analysis, or whether the same cells sometimes end up in one population and sometimes in a totally different one.

On the other hand, if the clusters shown are identical (i.e. cells belonging to a specific cluster in Seurat maintain that cluster identity in Monocle), then exactly the same name and number of clusters as well as their individual colors must be used, and this clearly stated in the manuscript.

This will make it possible to i) make sure how well the two clustering efforts overlap with each other and ii) show how cells that belong to the two PreBCRi II clusters divide in Monocle based on them belonging to different parts of the cell cycle.

3) Not absolutely necessary, but an improvement, would be a comparison between different ways of labeling bone marrow cells as B cells and to get some information which cell types are missing from the analysis. Simply, from a number of mice, label bone marrow cells with antibodies against CD19, CD79a as well as the ones used for sorting (B220, CD11c, CD43) and try to give a number what proportion of B cells that were lost due to the sorting antibodies used.

Response to Review – MS NCOMMS-21-02756A

Reviewer 1 was satisfied with the revisions to our manuscript. However, Reviewers 2 and 3 still wanted additional information. Our response to their critique is outlined below in red text. Changes to the revised manuscript are highlighted in yellow.

Reviewer #1 (Remarks to the Author):

The authors have added new supplementary figures and revised the manuscript - first, these revisions have improved the paper and second, they are responsive to the reviewers. Further revisions are not needed.

This paper reports very interesting results and is a nice step forward for this field

We are glad the reviewer was satisfied with our revisions.

Reviewer #2 (Remarks to the Author):

The authors have made significant efforts to improve their manuscript and have addressed most of the comments well. However, some of the comments have not been tackled properly and need to be improved before publication:

- The authors make no comments in the manuscript about how doublets were removed and this should be included in the methods. In the response to reviewer 1, we commend the authors for using the hashtag-seq information to identify doublets, however, this should be clearly described in the methods, such as how they defined positivity for each sample. The other thing to note is that this will only identify cross-sample doublets, which the authors claim to be only 83% of doublets. Therefore, I recommend another approach to identify these (MLtiplet, DoubletFinder or equivalent may be helpful). This must be addressed.

We have provided a more detailed description of how doublets were identified in the methods section (lines 636-655). We believe 83% removal of doublets is actually pretty good and that hashtagging is a more accurate way to identify doublets than other alternatives. However, as suggested by the reviewer we considered using both MLtiplet and DoubletFinder algorithms to identify doublets. The MLtiplet approach is actually very similar to the hashtagging approach we used as it relies on CITE-Seq antibodies that have distinct expression profiles to identify doublets based on dual antibody expressing cells. For example, in a sample containing B cells and T cells we would expect very distinct expression patterns for CD19 and CD3 CITE-Seq antibodies; cells expressing both markers would be considered doublets by MLtiplet. Unfortunately, the CITE-Seq antibodies used in our experiment (CD43, CD25, CD93, B220, CD19 and IgM) do not exhibit such stark all-or-nothing expression patterns as one moves from one developmental stage to the next and are thus not useful for the MLtiplet approach. MLtiplet can also use BCR sequence data to identify doublets; however, since we used 10xGenomics 3' scRNA-Seq kit for our studies BCR data is not available. Thus, MLtiplet is not really any better than the hashtagging based approach we already used.

We also tried to use DoubletFinder. The DoubletFinder algorithm tries to identify distinct gene signatures associated with single cells. It then computationally merges these signals to try and establish a gene signature for computationally predicted doublets. Once again this works very well in a situation where cell types are very distinct (i.e., a mixture of B cells and T cells). The authors of the study describing DoubletFinder acknowledge that DoubletFinder is not very good at finding homotypic doublets (for example, a doublet consisting of two mature B cells). We believe that Doublet Finder is also not optimal for studies examining developmental progressions, as by definition, as cells transition from one transcriptional state to another (i.e., pro-B cell to pre-B cell) they will pass through a transcriptional intermediate state.

Nevertheless, we did employ DoubletFinder and compared those results against our previous HTODemux results (i.e. Seurat's doublet classification function). Interestingly, DoubletFinder classified 893 WT GEMs as doublets that were called singlets by HTODemux. These 893 "potential" doublets were evenly dispersed across the entire dataset. We removed these cells and reprocessed the entire analysis. We recovered largely the same exact clusters. The main exception involves dividing B cells (especially cycling immature B cells), which are preferentially labeled as doublets. This cell cluster would contain signatures associated with cell cycle and differentiation, which might be particularly difficult for DoubletFinder to accurately classify. More importantly, DoubletFinder also classified 1,576 GEMs as singlets, when they clearly expressed multiple different HTOs in the same GEM as detected by HTODemux. This significant discrepancy calls into question the general accuracy of DoubletFinder with our dataset. The results from the DoubletFinder analysis are shown below. As can be seen, even after

A. Overlap between singlets and doublets identified by DoubletFinder or Seurat HTO DeMux algorithms in biological replicates. Most Singlets and doublets were identified by both DoubletFinder and DeMux (first two columns of each panel) but there were discrepancies. **B.** Distribution of Doublets identified by Doublet Finder projected onto original UMAP plot. **C.** Left plot is distribution of just doublets identified by Doublet Finder; right plot is same UMAP with all doublets identified by HTO DeMux and DoubletFinder removed.

removing ostensible doublets using DoubletFinder, we get a very similar distribution of cell types. We appreciate the suggestion to use alternative doublet identification methods in addition to HTO tags. However, since our data do not appear to change significantly upon applying DoubletFinder, and since we believe there are significant problems associated with applying this algorithm to cells in a developmental progression, we would prefer not to show what we believe could be a misleading analysis (although the data shown above will of course be available in the associated peer review file).

- Following up from reviewer 1's comments, it would be useful to have a figure of the CITE-seq and GEX signatures of each population so that the readers can be convinced of the groupings (such as a dotplot).

We think much of this information is already present in the data in the feature plots presented in figure 1. We are not sure if adding this in another format will add much, but could do so if the reviewer and editors feel it is absolutely necessary.

- Figure 7 is still not very informative for a reader. Can the authors indicate which modules are statistically associated with the different leukaemia subtypes? Otherwise, a reader cannot conclude that this is true signal or driven by noise.

We have reanalyzed this data and now include a statistical test showing that the two main clusters of leukemias are statistically distinct. Specifically, we found that many of the more aggressive leukemia subtypes are enriched for a proliferating pro/preB cell gene signature. In contrast, most of the less aggressive leukemias cluster with the pre-proB or immature/mature B cell gene signatures. This statistical analysis is now included in figure 7B and described on lines 308-316.

- Line 292: "Finally, we examined whether YBX3 expression correlated with outcome in B-ALL": have the authors corrected for multiple testing here? Presumably the authors tested all of the "pre-BCR module genes, including YBX3 and NRG1, [that] are significantly upregulated in BCL2/MYC, IKZF1 N159Y, and MEF2D B-ALL subtypes", and therefore just showing the results for YBX3 may be misleading. Were any of the other genes significantly associated? This needs to be addressed before any conclusions can be made in the manuscript.

The YBX3 data was obtained before any of the clustering analysis was done – it was the only gene we tested. We were prompted to do so based on its very high expression in the pre-BCRd cluster (see figure 3A). Unfortunately, the dataset we used to query survival associations only allows us to test individual genes (not gene sets), so were unable to do the analysis suggested here. Had we been able to do such additional analyses this would have required additional correction for multiple samples as requested.

- In the rebuttal to reviewers the authors provide the "The overlap for each cluster between Monocle and Seurat" (page 12), however some of the memberships for the Seurat list are very low (<0.05), suggesting poor agreement and low consistency between methods. This does not

seem appropriate, and wonder how the analysis would look if the authors correlated Seurat-only derived gene sets and monocle-only derived gene sets?

The reason for the small number is that monocle only uses the top 25 genes, while Seurat uses many more. Thus, if the Seurat list has 250 genes and 25 of them overlap with the monocle list this would imply only 10% overlap despite the fact that a higher result is impossible. We have done the same clustering analysis using either just the Seurat or just the monocle genes, as opposed to the intersection and obtained very similar results (see figure below). Only two leukemia subset (ETV6-Runx1 and hyperdiploid) switched groups based on the gene set used for clustering and they are the two that appear most borderline. We mention this as data not shown in the text, but could include it as a supplementary figure if the reviewer and editor felt this was critical (once again it will be available in the associated peer review file).

Left panel. Clustering done using Seurat-based gene set. Right panel. Clustering done using the monocle-derived gene set.

Reviewer #3 (Remarks to the Author):

My original concerns with this manuscript is listed below with a short description of the response from the authors

Original concern - author response

Only based on two mice from one experiment - the authors have not added additional experiments and argue that the experimental setup is sufficiently robust

No data on protein expression of markers identified as differentially expressed - the authors have added flow cytometry data that support their findings.

Data presented as functional data do not really support their claims – the authors have tried to address if there is redundancy between YBX1 and YBX3 but failed to delete both simultaneously in a B cell line.

The model suggested by the authors would require cells to lack B cells in G1 – the authors links to a pre-print studying proliferation of mature B cells that suggest that B cells may express some G2/M genes even during G1

The potential that distinct B cells subtypes may be lacking due to using B220 and CD11c for cell sorting – this is mentioned in the discussion

The number of doublets seem to be very high compared to what is expected – the authors show that the experiment included more hashtagged samples than the two presented. This explains the inconsistency since doublets counted include all doublets, including those between cells in this analysis and other cells.

Are the clusters in Monocle the same ones as the ones in Seurat? – the authors do not really address this in their responses and are still vague whether cells belonging to a specific Seurat cluster will be kept in that specific cluster in Monocle (i.e. if a specific cell that belonged to a specific Seurat cluster will it keep its color in Monocle), or if the clustering in Monocle of cells is different from the one used in Seurat (i.e. that the cells were reclustered and recolored in Monocle using a similar algorithm as in Seurat but with cell cycles regressed out so that some cells may belong to different clusters in Seurat and Monocle).

The B-ALL analysis is not described in any detail – the authors have improved this significantly in the revised manuscript.

In most cases I think that the authors have addressed my concerns, and that the manuscript have improved in the process.

However, I still have some concerns that I think must be further addressed before publication

Specifically

1) The lack of a G1 phase or not is very important for the conclusion in the manuscript. Many conclusions in the manuscript will change if the cycling cells to the right in the UMAP plot (Cycling ProB, Pro B VDJ, PreBCRd, PreBCRi) are connected to non-cycling or G1 cells ProB VDJ or Kappa Pre B) in the same cell cycle or not. In fact, if they are, this would not really support the model for early B cell differentiation suggested by the authors. I agree with the authors that the preprint to some extent could explain why it is hard to find typical G1 stage B cells among cycling B cells. However, while the preprint suggests some overlap between genes expressed in G2/M and G1 in cycling B cells, possibly due to that rapidly dividing lymphocytes not need additional stimulatory signals in G1 to continue to cycle, this does not mean that cells

do not enter G1 phase at all. In fact, as many as 70% of the B cells were classical G1 cells (based on DNA content and DNA synthesis) in the preprint. Rather, the major finding in that report is that there are cells mostly in G1 that express some G2/M phase genes.

I fully appreciate the work that the authors already have put into the analysis of the data in the current manuscript, but it should be relatively easy to address whether individual cells express both G1 and G2/M genes when proliferating or only G2/M genes. At the very least, the authors should look for typical G1 genes among their G2/M class cells at a single cell level and also the combined expression of genes included in their cell cycle analysis (Figure 1C) - do some cells have high values for both G1 and G2M genes within the preBCR clusters? In addition, by isolating the individual cell clusters and then sub-cluster them, possibly while regressing out cell cycle genes, may allow for a higher resolution of the analysis.

I do not find that it is justified to present the current data based on the assumption that bone marrow cells would not have any distinguishable G1 phase - which in fact would be a very major finding. We have found that cell cycle gene expression differences are often dominant when single-cell RNASeq data from cycling cells are clustered, and that this may sometimes hide true differences. I do not think that the authors sufficiently prove that this is not the reason for that some of the different subpopulations presented here are found. This can, and should, be addressed from existing data.

This is an important point and one that we believe we have finally been able to better resolve. We believe the main problem is largely due to how the Seurat program calls G1 versus S and G2/M cells. Essentially, this algorithm identifies cells expressing S phase genes or G2M phase genes and designates whatever is left as G1. However, if we plot some canonical G1 versus S and G2M genes it is quite clear that they are differentially expressed. We now present some of this data in a revised figure 1C and in supplemental figure 2A. Specifically, we calculated a gene expression score for the newly described G1 post-mitotic gene set (derived from the BiorXiv paper). All three cell cycle scores (i.e. an average expression value for S, G2M, or G1PM gene sets) are presented in figure 1C and supplemental figure 2A. A description of the G1PM genes, and how our analysis was done, is provided in the methods (lines 681-686). The G1PM genes are upregulated specifically in lymphocytes after undergoing multiple rounds of the cell cycle (described in the BiorXiv publication). Interestingly, in our experiment, the cells expressing the G1PM gene set most significantly are found at the interface between the G2M and S phase cells. Therefore, we contend that the cells in the proliferating stages are cycling through all phases of the cell cycle. Based on this analysis, we have changed the way we classify each cell by phase. Previously, if a cell was not expressing S or G2M phase genes at a high level, it was classified as G1. In our updated Figure 1C, we have added the G1PM classification to distinguish between cells in a relatively quiescent G0/G1 phase versus cells in the G1 post-mitotic phase rapidly moving around the cell cycle. This hopefully provides some nuance to the cell cycle story.

2) The correlation between clusters in the Seurat and Monocle analysis must be made absolutely clear.

Two of the Seurat clusters (PreBCRi II S phase and PreBCRi II G2/M phase) are presented as one cluster in Monocle (PreBCRi II), and Immature B and Cycling Immature (Seurat) presented as Immature B I and Immature B II (Monocle). Furthermore, statements such as "Using stage-defining markers, we identified 13 distinct B cell developmental stages" and "After adjustment for cell cycle status, UMAP dimensional reduction and clustering were performed in Monocle" suggest that the clusters may not be identical.

If the cells are reclustered in Monocle, after regressing out cell cycle genes, then this must also be made absolutely clear in the text. In addition, and the authors need to present the Monocle dot-plot representation with cells identities based on the Seurat clustering as a Supplementary figure to allow for the reader to compare how well the two methods overlapped with each other in identifying identically named clusters. Without this information it is impossible to determine if there is any correlation between the two methods for analysis, or whether the same cells sometimes end up in one population and sometimes in a totally different one.

On the other hand, if the clusters shown are identical (i.e. cells belonging to a specific cluster in Seurat maintain that cluster identity in Monocle), then exactly the same name and number of clusters as well as their individual colors must be used, and this clearly stated in the manuscript.

This will make it possible to i) make sure how well the two clustering efforts overlap with each other and ii) show how cells that belong to the two PreBCRi II clusters divide in Monocle based on them belonging to different parts of the cell cycle.

We have now repeated this analysis and cross-correlated the clusters derived from the Seurat and monocle-based analysis. In general, we find that there is very good concordance – cells that are found in one cluster in Seurat are typically found in the same cell cluster using Monocle. This data is shown visually in a revised figure 6A and in detail in supplementary Table 3. The main exceptions would be the case where one cluster is split in two. For example, the preBCRi II S cluster and G2/M cluster in Seurat were merged into PreBCRi II in Monocle. Conversely, the immature B cell cluster in Seurat was split into two immature B cell clusters in Monocle.

3) Not absolutely necessary, but an improvement, would be a comparison between different ways of labeling bone marrow cells as B cells and to get some information which cell types are missing from the analysis. Simply, from a number of mice, label bone marrow cells with antibodies against CD19, CD79a as well as the ones used for sorting (B220, CD11c, CD43) and try to give a number what proportion of B cells that were lost due to the sorting antibodies used.

We have run this analysis as requested. The vast majority of B cells in the bone marrow (>97%) are captured by our sort scheme, although there is a small subset of CD19+ cells expressing GR1, CD11c or NK1.1 that would be excluded. This new information is included in supplementary figure 1B and described in the text on lines 77-80).

REVIEWERS' COMMENTS

Reviewer #2 (Remarks to the Author):

The authors have added new analyses and revised the manuscript and have adequately responded to most of the reviewers' comments. I think this is an interesting paper that will benefit the field.

Reviewer #3 (Remarks to the Author):

I had two major concerns regarding the revised manuscript from the previous round.

i) That the authors were not able to define any cells in the G1 phase among rapidly proliferating cells

ii) That it was impossible to judge from the data how the subpopulations defined in Seurat corresponded to the ones defined in Monocle.

The authors have satisfactorily addressed these points in the current version of the manuscript by more carefully analysing and describing the single-cell data it is based on, and have additionally added experiment that demonstrate the proportion of cells that may be lost during flow cytometric sorting.

Overall, I think that these additions have improved the manuscript significantly, and that it is now suitable for publication.